# Mutant ASXL1 cooperates with BAP1 to promote myeloid leukaemogenesis

Shuhei Asada[1], Susumu Goyama[1], Daichi Inoue[1,2], Shiori Shikata[1], Reina Takeda[1], Tsuyoshi Fukushima[1], Taishi Yonezawa[1], Takeshi Fujino[1], Yasutaka Hayashi[1], Kimihito Cojin Kawabata[1,3], Tomofusa Fukuyama[1], Yosuke Tanaka[1], Akihiko Yokoyama[4], Satoshi Yamazaki[5], Hiroko Kozuka-Hata[6], Masaaki Oyama[6], Shinya Kojima[7], Masahito Kawazu [8], Hiroyuki Mano[7,9] & Toshio Kitamura[1]

*ASXL1* mutations occur frequently in myeloid neoplasms and are associated with poor prognosis. However, the mechanisms by which mutant ASXL1 induces leukaemogenesis remain unclear. In this study, we report mutually reinforcing effects between a C-terminally truncated form of mutant ASXL1 (ASXL1-MT) and BAP1 in promoting myeloid leukaemogenesis. BAP1 expression results in increased monoubiquitination of ASXL1-MT, which in turn increases the catalytic function of BAP1. This hyperactive ASXL1-MT/BAP1 complex promotes aberrant myeloid differentiation of haematopoietic progenitor cells and accelerates RUNX1-ETO-driven leukaemogenesis. Mechanistically, this complex induces upregulation of posterior *HOXA* genes and *IRF8* through removal of H2AK119 ubiquitination. Importantly, BAP1 depletion inhibits posterior *HOXA* gene expression and leukaemogenicity of ASXL1-MT-expressing myeloid leukemia cells. Furthermore, BAP1 is also required for the growth of MLL-fusion leukemia cells with posterior *HOXA* gene dysregulation. These data indicate that BAP1, which has long been considered a tumor suppressor, in fact plays tumor-promoting roles in myeloid neoplasms.

[1] Division of Cellular Therapy, Advanced Clinical Research Center, and Division of Stem Cell Signaling, Center for Stem Cell Biology and Regenerative Medicine, Institute of Medical Science, The University of Tokyo, Tokyo 1088639, Japan. [2] Human Oncology and Pathogenesis Program, Memorial Sloan Kettering Cancer Center, New York, NY 10065, USA. [3] Department of Hematology/Oncology, Weill Cornell Medical College, New York, NY 10021, USA. [4] National Cancer Center Tsuruoka Metabolomics Laboratory, Yamagata 9970052, Japan. [5] Division of Stem Cell Therapy, The Institute of Medical Science, The University of Tokyo, Tokyo 1088639, Japan. [6] Medical Proteomics Laboratory, The Institute of Medical Science, The University of Tokyo, Tokyo 1088639, Japan. [7] Department of Cellular Signaling, Graduate School of Medicine, The University of Tokyo, Tokyo 1130033, Japan. [8] Department of Medical Genomics, Graduate School of Medicine, The University of Tokyo, Tokyo 1130033, Japan. [9] National Cancer Center Research Institute, Tokyo 1040045, Japan. Correspondence and requests for materials should be addressed to T.K.(email: kitamura@ims.u-tokyo.ac.jp)

Additional sex combs-like 1 (ASXL1) is a member of the ASXL family and is involved in epigenetic regulation[1, 2]. Mutations in the *ASXL1* gene are frequently found in myeloid neoplasms, including myelodysplastic syndromes (MDS), chronic myelomonocytic leukemia (CMML), and acute myeloid leukemia (AML)[3–8]. These mutations are predominantly frameshift and nonsense mutations generating C-terminally truncated proteins, and are associated with worse prognosis[8]. *ASXL1* mutations have also been implicated in clonal haematopoiesis of indeterminate potential, suggesting that it is among the earliest genetic events in the process of myeloid transformation[9–11].

Members of the ASXL family share a common domain architecture, which includes a highly conserved ASX homology (ASXH) domain at the N-terminal region and a plant home-odomain (PHD) finger at the C-terminal region[12]. It has been suggested that the PHD domain, which is lost in most *ASXL1* mutations, binds histones with specific modifications and recruits chromatin modulators and transcriptional factors[13]. The ASXH domain mediates interaction with a partner protein BAP1. BAP1 is an essential component of the "polycomb repressive deubiquinase complex (PR-DUB)," in which it deubiquitinates monoubiquitinated histone H2A at lysine 119 (H2AK119ub), a modification that is catalyzed by the polycomb repressive complex 1 (PRC1)[14]. The mammalian PR-DUB complex contains ASXL family proteins, which are required for its deubiquinating activity[15]. In addition to BAP1, ASXL1 interacts directly with EZH2, EED, and SUZ12, catalytic and scaffold subunits of PRC2, which promotes trimethylation of H3 at lysine 27 (H3K27me3)[16, 17]. Thus, ASXL1 may act as an epigenetic scaffold in the regulation of various histone modifications, including H2AK119ub and H3K27me3.

How ASXL1 mutations induce myeloid transformation is not fully understood. Previous studies have reported that ASXL1 knockdown and genetic deletion of *Asxl1* in haematopoietic cells promotes myeloid transformation[12, 16, 18], indicating that mutations in ASXL1 produce loss of function. However, a growing body of evidence suggests that *ASXL1* mutations in fact result in gain of function. Experiments using mouse bone marrow transplantation models have revealed that forced expression of a C-terminally truncated ASXL1 mutant in haematopoietic progenitor cells induces MDS-like diseases, and accelerates AML development in concert with Nras or SETBP1 mutations[17, 19]. In patients with *ASXL1* mutations, the mutations are typically heterozygous and occur near the 5′ end of exon 12, thus producing C-terminally truncated forms of ASXL1 probably escaping from nonsense-mediated decay (NMD) of mRNA, and indeed truncated ASXL1 proteins are expressed in MDS cells[20]. Thus, whether ASXL1 mutations promote myeloid transformation via a gain or loss of function remains an unresolved question.

Mechanistically, it has been shown that both *Asxl1* deletion and mutant *Asxl1* overexpression induce global reduction of H3K27me3 in haematopoietic cells[12, 16–18]. These data suggest that loss of ASXL1 function in promoting H3K27me3 contributes to myeloid transformation. On the other hand, recent studies have shown that cancer-associated ASXL1 mutations aberrantly enhance BAP1 function in the deubiquination of H2AK119ub, raising the possibility that increased PR-DUB activity underlies the oncogenic effect of *ASXL1* mutation[15, 21]. However, the precise nature of the epigenetic dysregulation, which plays a major role in mutant ASXL1-induced leukaemogenesis, remains unknown.

In the present study, we report a mutually reinforcing effect between mutant ASXL1 and BAP1, which promotes myeloid leukaemogenesis. BAP1 induces stabilization and monoubiquitination of mutant ASXL1, and monoubiquitinated ASXL1-MT increases the catalytic function of BAP1. This hyperactive mutant ASXL1/BAP1 complex induces upregulation of posterior *HOXA* genes and *IRF8* through inhibition of H2AK119ub, impairing multilineage differentiation of haemato-poietic progenitors (except for that toward monocytes), and accelerates RUNX1-ETO-induced leukaemogenesis. Importantly, Bap1 depletion using CRISPR/Cas9 substantially inhibits the leukaemogenicity of myeloid leukemia cells expressing mutant ASXL1. BAP1 is also required for the growth of MLL-fusion leukemia cells through the upregulation of *HOXA* gene expression. These data indicate that BAP1, which has long been regarded as a beneficial tumor suppressor, also plays a tumor-promoting role in myeloid leukaemogenesis.

## Results

**BAP1 induces monoubiquitination of mutant ASXL1**. We first examined the interaction between a leukemia-associated ASXL1 mutant [ASXL1 (1900–1922del; E635RfsX15[17], which here we refer to as ASXL1-MT]) and BAP1 in 293T cells (Fig. 1a). Coexpression of BAP1 increased expression of ASXL1-MT, and interestingly, also caused a distinct mobility shift of ASXL1-MT (Fig. 1b). This BAP1-induced mobility shift was also observed when we used the other leukemia-associated ASXL1 mutations [(c.1934dupG; G646WfsX12 and c.1772dupA; Y591X)] (Supplementary Fig. 1a). Importantly, this effect was absent when we used ASXL1-MT-K351R, a ubiquitination-deficient form of ASXL1-MT[22] (Fig. 1b). A catalytically inactive BAP1 mutant, BAP1-C91S, also caused a mobility shift in ASXL1-MT, but showed weaker activity in upregulating ASXL1-MT expression (Fig. 1b). In a cycloheximide (CHX)-chase experiment, turnover of ASXL1-MT protein with BAP1 coexpression was slower than that of ASXL1-MT alone, indicating that BAP1 upregulates ASXL1-MT expression by enhancing its stability (Fig. 1c). BAP1 also increased expression of wild-type ASXL1, although to a lesser extent (Fig. 1d). Consistent with this observation, BAP1 had a higher affinity for ASXL1-MT than for wild-type ASXL1 (Supplementary Fig. 2a, b). In immunoprecipitation assays, BAP1 interacted with ASXL1-MT and induced its monoubiquitination. BAP1 also interacted with ASXL1-MT-K351R and upregulated its expression, but did not induce monobubiquitination of ASXL1-MT-K351R (Fig. 1e and Supplementary Figs. 1c, 2c, d). These data suggest that BAP1 stabilizes ASXL1-MT and induces its monoubiquitination at lysine 351.

Interestingly, we detected the slowly migrating protein (monoubiquitinated ASXL1-MT) in cells that were retrovirally transduced with ASXL1-MT without BAP1 overexpression (Supplementary Fig. 1b), while such mobility shift was observed only with BAP1 overexpression in a transient transfection assay. Because retroviral transduction produced relatively low level of ASXL1-MT than the transient transfection, we speculated that relative expression of ASXL1-MT and BAP1 determines the ratio of mono- and non-ubiquitinated ASXL1-MT. To test this hypothesis, we sorted GFP-high and -low fraction in 293T cells that were retrovirally transduced with ASXL1-MT (marked with GFP), and assessed ubiquitination status of ASXL1-MT in each fraction (Fig. 1f). The majority of ASXL1-MT was detected as the monoubiquitinated protein in GFP-low fraction (expressing low level of ASXL1-MT), while both mono- and non-ubiquitinated proteins were detected in GFP-high fraction (expressing high level of ASXL1-MT) (Fig. 1g). These data support our hypothesis, and suggest that endogenous ASXL1 mutants probably exist mainly as the monoubiquinated forms in cells. To assess the role of endogenous BAP1 for monoubiquitination of ASXL1-MT, we then transduced Cas9 into ASXL1-MT-expressing 293T and Hela

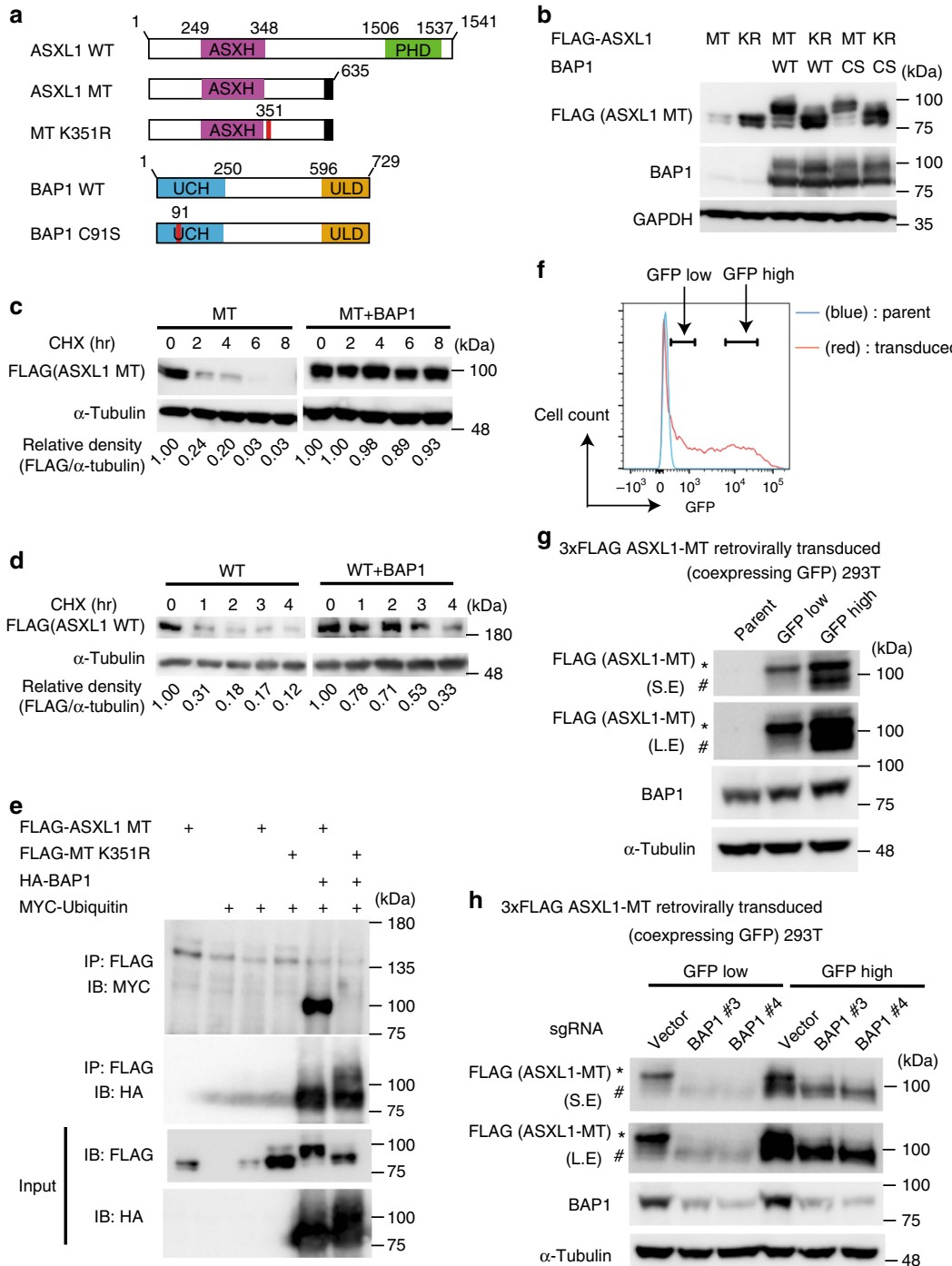

**Fig. 1** Expression of BAP1 stabilizes ASXL1-MT and induces its monoubiquitination at lysine 351. **a** Schematic presentation of ASXL1 and BAP1 constructs. ASXH Asx homology domain, PHD PH domain, UCH ubiquitin C-terminal hydrolase, ULD Uch37-like domain. **b** 293T cells were transfected with FLAG-ASXL1-MT (MT), FLAG-ASXL1-MT-K351R (KR), BAP1, and BAP1-C91S (CS). Cell lysates were subjected to western blotting analysis. **c**, **d** 293T cells were transfected with FLAG-ASXL1-MT (**c**) or FLAG-ASXL1-WT (**d**) together with vector or HA-BAP1. Forty-eight hours later, cells were treated with 50 μg/ml cycloheximide (CHX) for the indicated times, and cell extracts were analyzed with anti-FLAG and anti-alpha-tubulin antibodies. The band intensities of FLAG relative to alpha-tubulin are shown. The value FLAG/alpha-tubulin without CHX treatment was set to 1. **e** 293T cells were transfected with FLAG-ASXL1-MT, FLAG-ASXL1-MT-K351R, HA-BAP1, and Myc-ubiquitin. Total cell lysates were immunoprecipitated with anti-FLAG M2 antibody, and the ubiquitinated ASXL1-MT was detected with anti-Myc. **f** 293T cells were retrovirally transduced with 3xFLAG-tagged ASXL1-MT (coexpressing GFP), and then GFP-high or -low fraction (expressing relatively high or low level of ASXL1-MT) was sorted. Shown is a FACS plot of gating strategy for GFP-low and GFP-high fractions. **g** Cell lysates extracted from parent, GFP-low (expressing low level of ASXL1-MT) and GFP-high (expressing high level of ASXL1-MT) 293T cells were subjected to western blotting analysis. (*): monoubiquitinated ASXL1-MT, (#): non-ubiquitinated ASXL1-MT. **h** GFP-low and GFP-high 293T cells were transduced with Cas9 together with vector or sgRNAs targeting BAP1. Cell lysates were subjected to western blotting analysis. (*): monoubiquitinated ASXL1-MT, (#): non-ubiquitinated ASXL1-MT. Experiments were independently repeated at least three times with similar results. Shown are representative results

cells together with sgRNAs targeting BAP1. As expected, BAP1 depletion resulted in the disappearance of monoubiquitinated ASXL1-MT in these cells (Fig. 1h and Supplementary Fig. 1d). Thus, endogenous BAP1 plays the essential role to induce monoubiquitination of ASXL1-MT.

**UBE2O is an E3 ubiquitin ligase for ASXL1-MT.** As BAP1 is a deubiquitinase, we predicted that it recruits ubiquitin ligase(s) to promote ubiquitination of ASXL1-MT. To identify such ligases, we performed nano liquid chromatography tandem mass spectrometry (nano-LC-MS/MS) using cell lysates from 293T cells transfected with ASXL1-MT alone or ASXL1-MT and BAP1 (Fig. 2a, b). This analysis confirmed the BAP1-induced ubiquitination of ASXL1-MT at lysine 351 (Supplementary Data 1, 2),

and revealed that interaction between ASXL1-MT and the atypical E2/E3 hybrid ligase UBE2O occurs only in BAP1-expressing cells (Supplementary Data 3, 4). This BAP1-mediated interaction between ASXL1-MT and UBE2O was confirmed by the immunoprecipitation assay (Supplementary Fig. 1e). Overexpression of UBE2O, but not its catalytically inactive mutant, promoted polyubiquitination of ASXL1-MT in 293T cells (Fig. 2c). Conversely, UBE2O depletion in 293T cells using the CRISPR/Cas9 system increased ASXL1-MT expression (Fig. 2d). However, UBE2O depletion did not inhibit the BAP1-induced mobility shift of ASXL1-MT, indicating involvement of other ligases promoting monoubiquitination of ASXL1-MT.

Next, we compared the effect of BAP1 and BAP1-C91S on the ubiquitination of ASXL1-MT. Immunoprecipitation assay

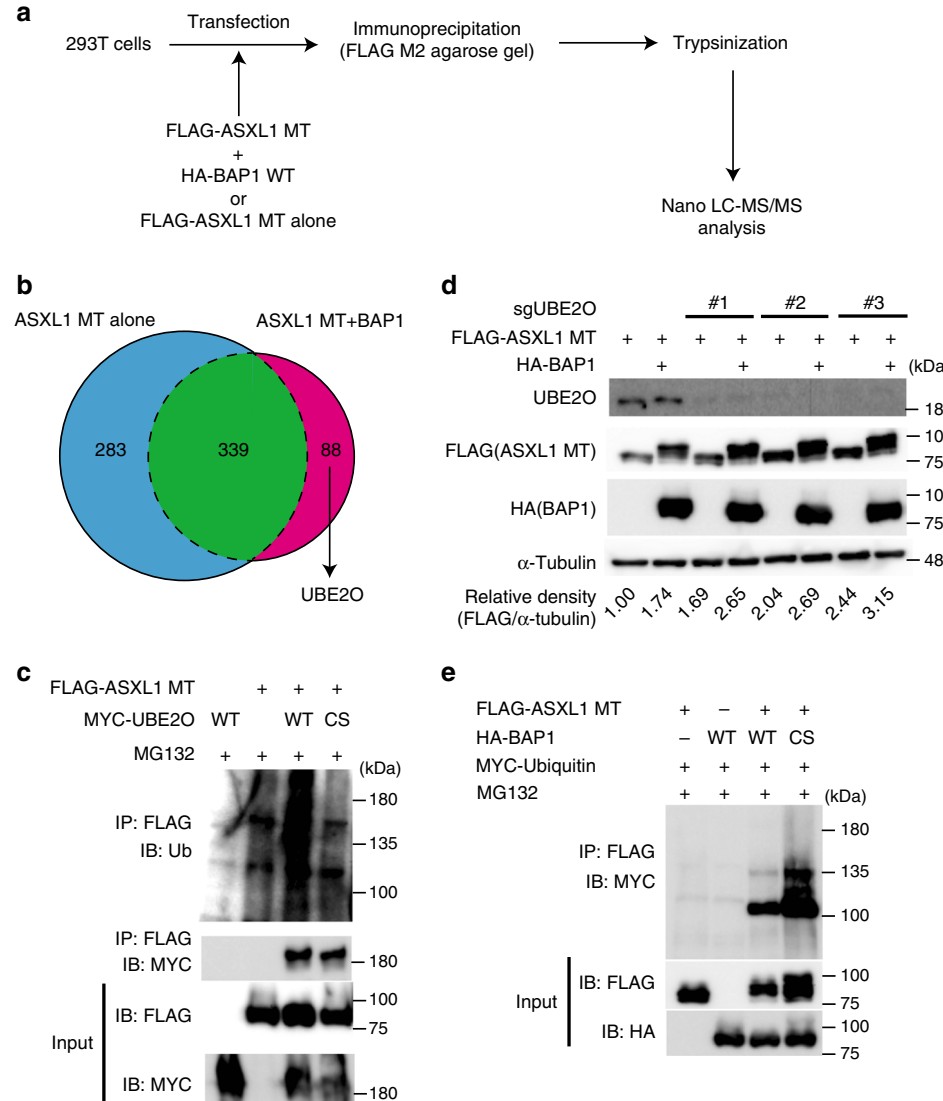

**Fig. 2** UBE2O is an E3 ubiquitin that induces polyubiquitination of ASXL1-MT. **a** Schematic experimental procedures of nano-LC-MS/MS analysis. **b** Venn diagram shows the overlap of proteins bound to ASXL1-MT in the conditions with or without coexpression of BAP1. Eighty-eight proteins, including UBE2O, bound to ASXL1-MT only with the coexpression of BAP1. See also Supplementary Data 3 and 4. **c** 293T cells were transfected FLAG-ASXL1-MT, MYC-UBE2O (WT), and MYC-UBE2O-C885S (CS). Total cell lysates were immunoprecipitated with anti-FLAG M2 antibody, and the ubiquitinated ASXL1-MT was detected with anti-ubiquitin antibody. **d** 293T cells were transduced with Cas9 and a vector control or three independent sgRNAs targeting UBE2O. Cells were then transfected with FLAG-ASXL1-MT and HA-BAP1. Cell lysates were subjected to western blotting analysis. The band intensities of FLAG relative to alpha-tubulin are shown. The value of FLAG/alpha-tubulin for ASXL1-MT alone without UBE2O depletion was set to 1. **e** 293T cells were transfected with FLAG-ASXL1-MT, HA-BAP1 (WT), HA-BAP1-C91S (CS), and Myc-ubiquitin. Total cell lysates were immunoprecipitated with anti-FLAG M2 antibody, and the ubiquitinated ASXL1-MT was detected with anti-Myc. Experiments shown in (**c–e**) were independently repeated at least three times with similar results. Shown are representative results

revealed that expression of BAP1-C91S efficiently induced both mono- and polyubiquitination of ASXL1-MT, while BAP1 induced only its monoubiquitination (Fig. 2e). Given that BAP1-C91S has reduced DUB activity, these results suggest that BAP1 recruits ubiquitin ligases including UBE2O to promote ASXL1-MT ubiquitination, and simultaneously removes poly-ubiquitin chain from it, leaving the monoubiquitinated form of ASXL1-MT. The decreased polyubiquitination of ASXL1-MT probably contributes to the increased stability of ASXL1-MT in BAP1-expressing cells, and explains why BAP1 increases ASXL1-MT expression more efficiently than BAP1-C91S. Thus, BAP1 enhances monoubiquitination of ASXL1-MT probably through the recruitment of ubiquitin ligases and the concurrent removal of the polyubiquitin chain from it.

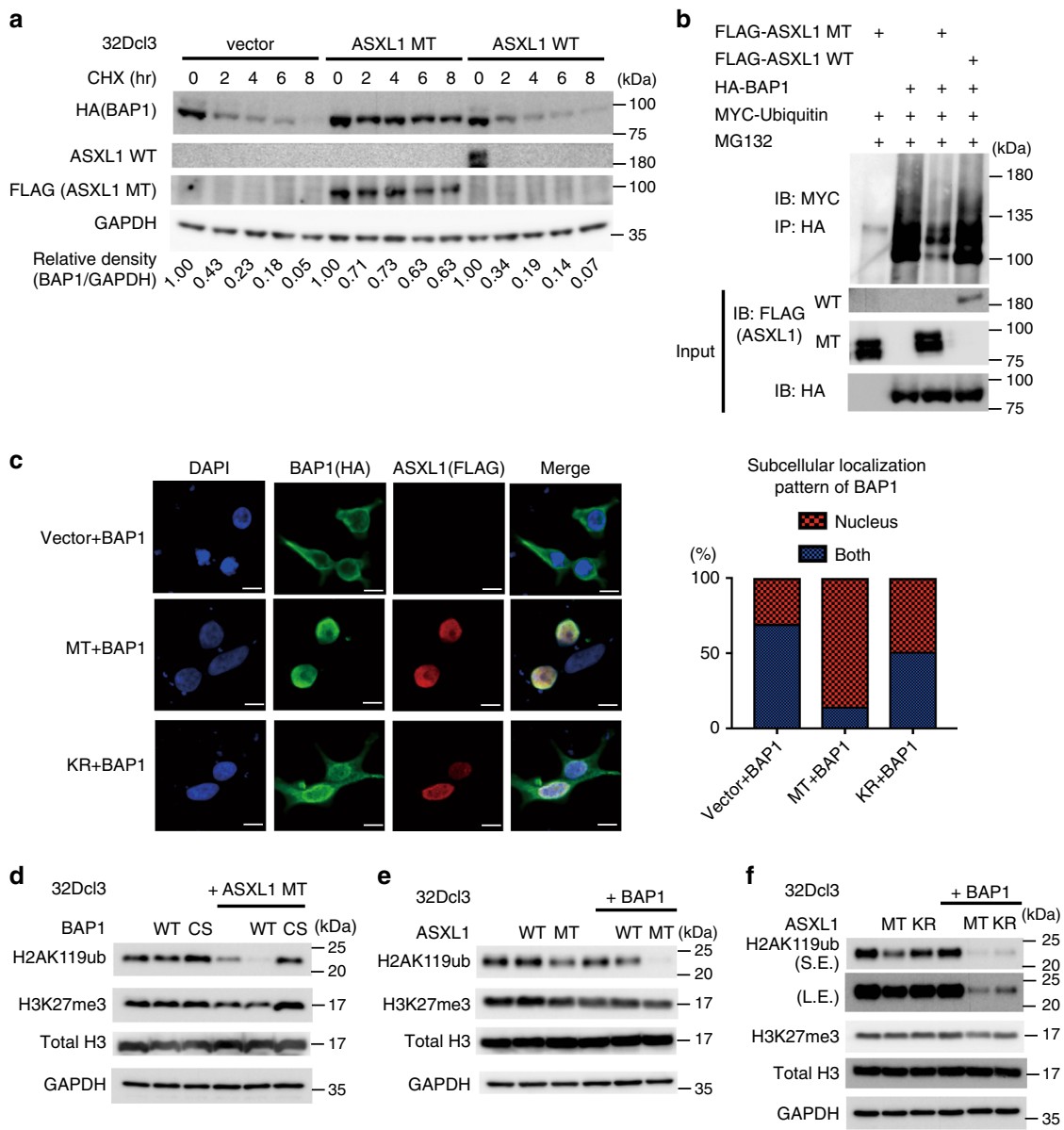

**Fig. 3** Monoubiquitinated ASXL1-MT enhances catalytic function of BAP1. **a** 32Dcl3 cells stably expressing HA-BAP1 together with vector, FLAG-ASXL1-MT, or FLAG-ASXL1-WT were treated with 50 μg/ml CHX for the indicated times, and cell extracts were analyzed with anti-HA, anti-FLAG, and anti-GAPDH antibodies. The band intensities of BAP1 relative to GAPDH are shown. The value of BAP1/GAPDH without CHX treatment was set to 1. ASXL1-MT, but not ASXL1-WT, stabilized BAP1. **b** 293T cells were transfected with FLAG-ASXL1-WT, FLAG-ASXL1-MT, HA-BAP1, and Myc-ubiquitin. Total cell lysates were immunoprecipitated with anti-HA antibody, and the ubiquitinated BAP1 was detected with anti-Myc. ASXL1-MT, but not ASXL1-WT, reduced ubiquitination of BAP1. **c** 293T cells were transfected with HA-BAP1 together with vector, FLAG-ASXL1-MT (MT), or FLAG-ASXL1-MT-K351R (KR), and were stained with anti-FLAG (rabbit) or anti-HA (mouse) antibody followed by secondary anti-rabbit Alexa 568 (red) or anti-mouse Alexa 488 (green) staining. Nuclei were visualized with DAPI (Blue). Confocal laser scanning microscopy (Nikon A1) was used to observe localization of ASXL1-MT and BAP1 (left). Scale bars: 10 μm. Subcellular localization of BAP1 was quantified by counting 400 cells exhibiting nuclear localization (nucleus) or diffuse distribution in both nucleus and cytoplasm (both). **d–f** 32Dcl3 cells were transduced with the combinations of wild-type ASXL1 (ASXL1-WT), ASXL1-MT (MT), ASXL1-MT-K351R (KR), BAP1 (BAP1-WT), and BAP1-C91S (CS). Cell lysates extracted from them were subjected to immunoblotting with anti-H2AK119ub, anti-H3K27me3, anti-total H3, and anti-GAPDH antibodies. Expression of ASXL1-MT (but not ASXL1-WT) and wild-type BAP1 (but not BAP1-C91S) strongly reduced H2AK119ub (**d**, **e**). ASXL1-MT-K351R showed attenuated activity to enhance BAP1-induced deubiquitination of H2AK119 (**f**). Experiments were independently repeated at least three times with similar results. Shown are representative results

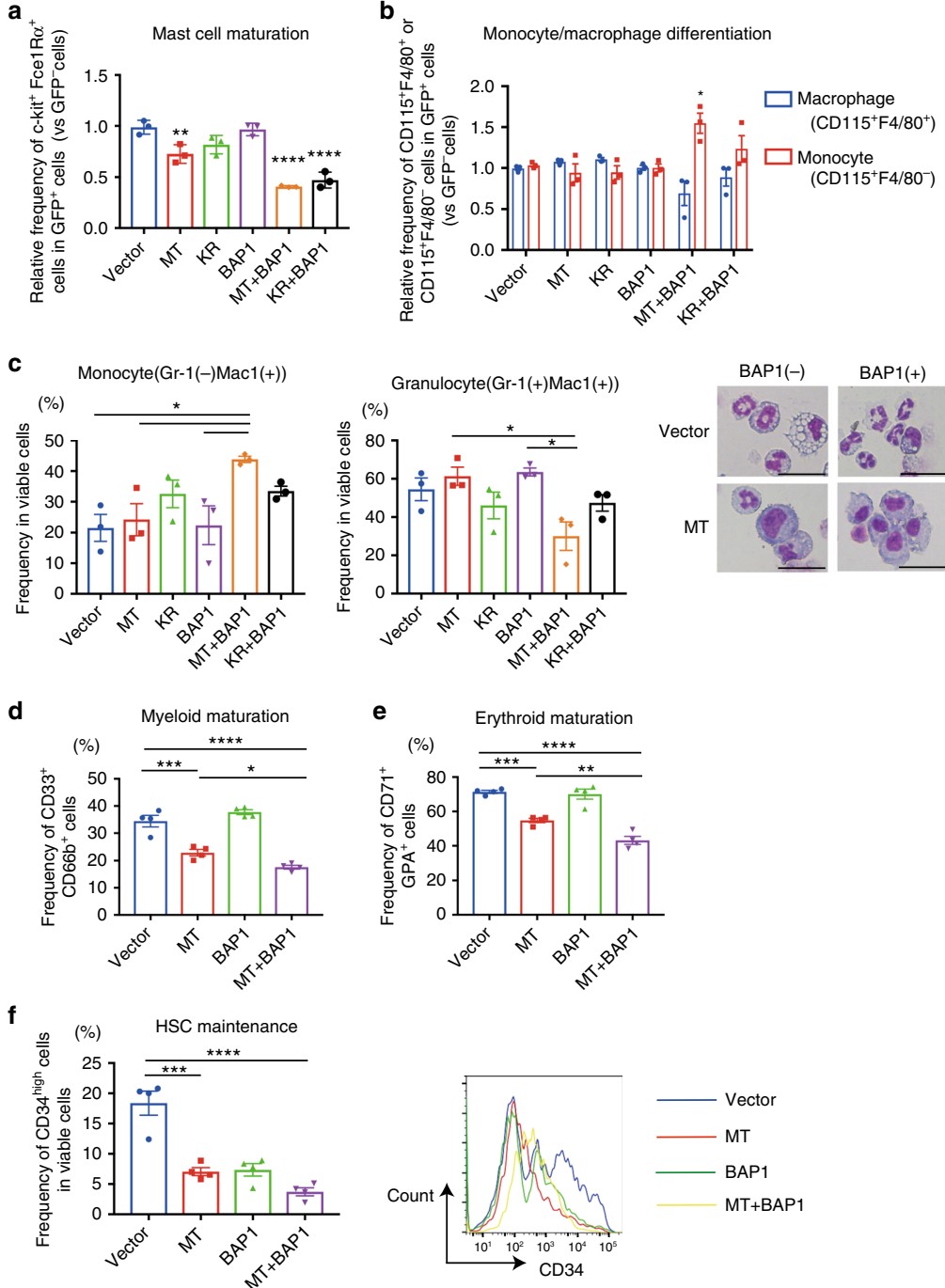

**Fig. 4** ASXL1-MT/BAP1 complex impairs multilineage differentiation of haematopoietic progenitors except for differentiation towards monocytes. **a**, **b** Murine c-Kit+ bone marrow cells were transduced with vector, ASXL1-MT (MT), or ASXL1-MT-K351R (KR) (coexpressing GFP) together with vector or BAP1 (coexpressing puromycin-resistant gene). After the puromycin selection for 48 h, cells were cultured with cytokines to induce mast cell or monocyte/macrophage differentiation. Mast cell maturation was assessed by the ratio of FcεIRa and c-Kit double-positive cells (enriched with mature mast cells) after 9 days of culture (**a**). Monocyte (CD115+F4/80−) and macrophage (CD115+F4/80+) differentiation was assessed after 7 days of culture (**b**). (n = 3 each) (**c**) Murine c-Kit+ bone marrow cells were transduced with vector, ASXL1-MT (MT), or ASXL1-MT-K351R (KR) (coexpressing blasticidin-resistant gene) together with vector or BAP1 (coexpressing puromycin-resistant gene). After the selection with puromycin and blasticidin for 3 days, cells were cultured with cytokines to induce differentiation toward both granulocytes and monocytes. Granulocyte (Gr-1+CD11b+) and monocyte (Gr-1−CD11b+) differentiation was assessed by FACS analyses (left) and Wright–Giemsa staining (right, scale bars: 20 μm) after 5 days of culture. (n = 3) (**d**–**f**) Human CB CD34+ cells were transduced with vector, ASXL1-MT (MT), or ASXL1-MT-K351R (KR) (coexpressing GFP) together with vector or BAP1 (coexpressing puromycin-resistant gene). After the puromycin selection for 48 h, cells were cultured in myeloid skewing (**d**), erythroid-skewing (**e**), and stem cell-maintaining (**f**) mediums, respectively. Myeloid maturation of CB cells was assessed by the ratio of CD33+CD66b+ cells after 7 days of culture (**d**). Erythroid maturation of CB cells was assessed by the ratio of CD79+CD235a+ cells after 5 days of culture (**e**). HSC maintenance of CB cells was assessed by the ratio of CD34-high cells after 7 days of culture (**f**). (n = 4 each) Data are shown as mean ± s.e.m. *P < 0.05, **P < 0.01, ***P < 0.001, ****P < 0.0001, one-way ANOVA with Tukey's multiple comparisons test

**Monoubiquitinated ASXL1-MT enhances BAP1 function**. Coexpression and immunoprecipitation assays using ASXL1-MT and BAP1 revealed that the presence of ASXL1-MT, but not ASXL1-WT, increased the stability of BAP1 and reduced its polyubiquitination, indicating that ASXL1-MT enhances auto-deubiquitination activity of BAP1 (Fig. 3a, b). Because auto-deubiquitination of BAP1 was shown to promote its nuclear retention[23], we assessed the effect of ASXL1-MT on the sub-cellular localization of BAP1 using immunofluorescence analyses. BAP1 normally showed a diffuse localization pattern in both nucleus and cytoplasm. Overexpression of ASXL1-MT promoted nuclear import of BAP1, which was observed only weakly when we used the ubiquitination-deficient ASXL1-MT-K351R (Fig. 3c). The ASXL1-MT-induced nuclear import was markedly

attenuated for BAP1-C91S (Supplementary Fig. 3a), indicating that autodeubiquitination of BAP1 is important for its nuclear localization. Thus, the monoubiquitinated ASXL1-MT enhances autodeubiquitination and nuclear retention of BAP1.

Next, we assessed the cooperative effect of ASXL1-MT and BAP1 on histone modifications in a murine myeloid cell line 32Dcl3. 32Dcl3 cells expressing ASXL1-MT showed global reduction of H2AK119ub, which was dramatically enhanced by the coexpression of BAP1, but not by that of BAP1-C91S (Fig. 3d). Expression of ASXL1-WT showed no effect on global reduction of H2AK119ub, which was only slightly enhanced by the coexpression of BAP1 (Fig. 3e). Essentially, the same results were obtained with 293T cells (Supplementary Fig. 3b, c). ASXL1-MT-K351R also reduced the levels of H2AK119ub in cooperation

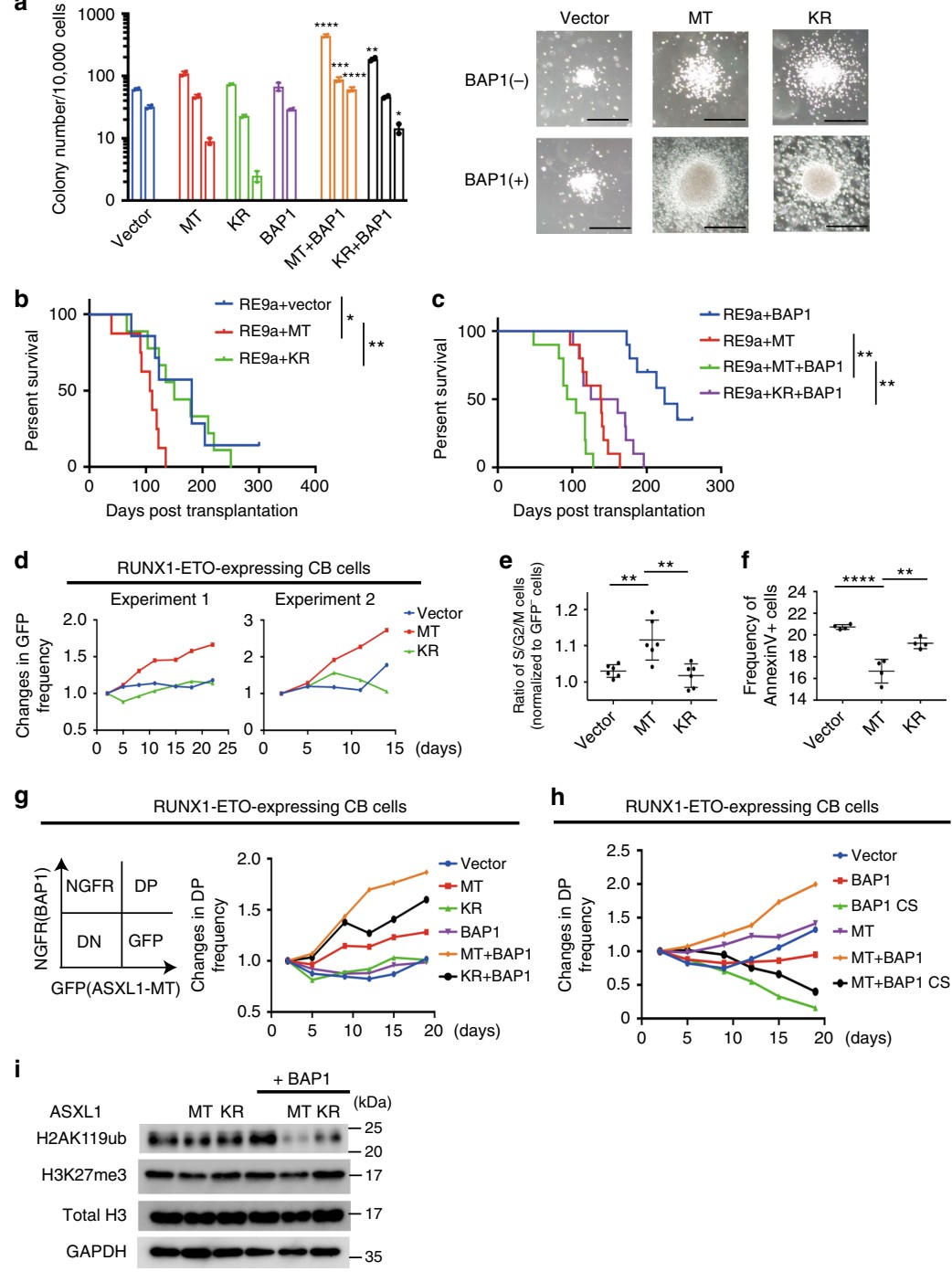

with BAP1, but this mutant consistently showed weaker activity compared with ASXL1-MT (Fig. 3f). Coexpression of ASXL1-MT and BAP1 also reduced bulk H3K27me3 levels, which is probably a direct consequence of loss of H2AK119Ub[24] in 32Dcl3 cells (Fig. 3d–f). Together, these data suggest that ASXL1-MT, especially the monoubiquitinated form of ASXL1-MT, increases catalytic function of BAP1.

**ASXL1-MT/BAP1 impairs multilineage differentiation of HSPCs.** Next, we assessed the effect of ASXL1-MT and BAP1 on multilineage differentiation of haematopoietic progenitors. We first used 32Dcl3 cells expressing ASXL1-MT and BAP1 to perform granulocytic differentiation assay. ASXL1-MT alone modestly, and coexpression of ASXL1-MT and BAP1 dramatically blocked G-CSF-induced granulocytic differentiation of 32Dcl3 (Supplementary Fig. 4a). We then transduced ASXL1-MT and BAP1 into murine c-Kit$^+$ progenitor cells, and induced myeloid differentiation with various combinations of cytokines[21, 25, 26]. In the mast cell differentiation assay, ASXL1-MT alone modestly impaired mast cell maturation of murine HSPCs, and this inhibitory effect was enhanced by BAP1 coexpression. ASXL1-MT-K351R showed similar but slightly reduced effects to produce mature mast cells compared with ASXL1-MT (Fig. 4a and Supplementary Fig. 4b). In the monocyte/macrophage differentiation assay, coexpression of ASXL1-MT and BAP1 substantially promoted differentiation toward CD115$^+$ monocytes while inhibiting their terminal differentiation to CD115$^+$F4/80$^+$ macrophages (Fig. 4b and Supplementary Fig. 4c). Again, coexpression of ASXL1-MT-K351R and BAP1 showed similar but weaker effects on monocyte/macrophage maturation. We also cultured the cells with SCF, IL-3, and IL-6 to induce differentiation toward both granulocytes and monocytes. This assay revealed that ASXL1-MT and BAP1 coexpressing cells favored monocyte differentiation, as evidenced by the substantial increase in the ratio of Gr-1$^-$CD11b$^+$ monocytes and decrease in the ratio of Gr-1$^+$CD11b$^+$ granulocytes (Fig. 4c).

We further assessed the role of ASXL1-MT and BAP1 in multilineage differentiation of human haematopoietic progenitors[27]. Human cord blood (CB) CD34$^+$ cells were transduced with ASXL1-MT or ASXL1-MT-K351R together with BAP1, and the cells were cultured with cytokines to induce myeloid or erythroid differentiation. ASXL1-MT expression decreased the ratio of CD33$^+$CD66b$^+$ granulocytes in the cultures designed to promote myeloid differentiation, and this effect was enhanced with BAP1 coexpression (Fig. 4d and Supplementary Fig. 4d). Similarly, expression of ASXL1-MT modestly, and coexpression of ASXL1-MT and BAP1 strongly decreased the ratio of CD71$^+$GPA$^+$ mature erythroid cells in the cultures designed to

promote erythroid differentiation (Fig. 4e and Supplementary Fig. 4e). The ASXL1-MT/BAP1-induced block of erythroid differentiation was also observed in TF-1 cells stimulated by erythropoietin (Supplementary Fig. 4f). Coexpression of ASXL1-MT and BAP1 decreased the ratio of CD34$^+$ cells when the cells were cultured with cytokines designed to promote maintenance of HSPCs, indicating that combination of ASXL1-MT and BAP1 did not simply increase the frequency of HSPCs (Fig. 4f). Together, these results indicate that the ASXL1-MT/BAP1 complex impairs terminal differentiation of HSPCs, except for differentiation toward monocytes.

**ASXL1-MT/BAP1 promotes RUNX1-ETO-induced leukaemogenesis.** To assess the role of ASXL1-MT and BAP1 in myeloid transformation, we transduced ASXL1-MT, ASXL1-MT-K351R, and/or BAP1 into murine c-Kit$^+$ progenitors and performed colony-replating assays (Fig. 5a). Cells expressing both ASXL1-MT and BAP1 produced more and larger colonies exhibiting higher levels of c-Kit expression than vector-transduced cells in the first three rounds. Coexpression of ASXL1-MT-K351R and BAP1 also increased the colony-forming activity of c-Kit$^+$ cells; however, similar to the earlier results, this ubiquitination-deficient form of ASXL1-MT consistently showed weaker activity than that of ASXL1-MT (Fig. 5a). Cells expressing ASXL1-MT and BAP1 did not produce colonies beyond the fourth passage, suggesting that the ASXL1-MT/BAP1 complex itself is not sufficient to transform murine HSPCs.

Next, we examined possible cooperation between ASXL1-MT and RUNX1-ETO (also called AML1-ETO) in promoting myeloid leukaemogenesis. RUNX1-ETO is a leukemia fusion protein and has been shown to increase self-renewal of HSPCs[28]. In addition, clinical data indicate that ASXL1 mutations and RUNX1-ETO frequently co-occur in AML patients[29, 30]. We transduced RUNX1-ETO9a, a shorter isoform of RUNX1-ETO with a stronger leukaemogenic activity than RUNX1-ETO, together with vector, ASXL1-MT or ASXL1-MT-K351R, into fetal liver cells, and transplanted these cells into sublethally irradiated recipient mice[31] (Supplementary Fig. 5a, b). Expression of ASXL1-MT, but not ASXL1-MT-K351R, accelerated the development of AML induced by RUNX1-ETO9a, suggesting an oncogenic role for the monoubiquitinated ASXL1 mutant (Fig. 5b and Supplementary Fig. 5c–e).

We then assessed the effect of coexpression of BAP1 and ASXL1-MT on the leukaemogenicity of RUNX1-ETO leukemia. After transducing fetal liver cells with RUNX1-ETO9a along with multiple combinations of ASXL1-MT, ASXL1-MT-K351R, and BAP1, we transplanted the cells into recipient mice. Coexpression of ASXL1-MT and BAP1 promoted RUNX1-ETO9a-induced

**Fig. 5** ASXL1-MT/BAP1 complex increases colony-replating activity of haematopoietic progenitors and promotes RUNX1-ETO-induced leukaemogenesis. **a** Bone marrow c-Kit$^+$ cells were transduced with vector/ASXL1-MT (MT)/ASXL1-MT-K351R (KR) (coexpressing blasticidin resistance gene) together with vector/BAP1 (coexpressing puromycin resistance gene). After the drug selection for 3 days, the cells were serially replated. Shown are weekly colony counts per 10$^4$ replated cells (mean ± s.e.m.) from duplicate plates (left, *$P < 0.05$, **$P < 0.01$, ***$P < 0.001$, ****$P < 0.0001$, one-way ANOVA with Tukey's multiple comparisons test), and representative photos of each colony at second round (right, Scale bars: 200 μm). **b** Fetal liver c-Kit$^+$ cells were transduced with RUNX1-ETO9a (RE9a) in combination with vector/MT/KR and were transplanted into recipient mice. Shown are the survival curves (vector: $n = 7$, MT: $n = 8$, KR: $n = 9$). *$P < 0.05$, **$P < 0.01$, log-rank test. **c** Fetal liver c-Kit$^+$ cells were transduced with RE9a, vector/MT/KR, and vector/BAP1, and were transplanted into recipient mice. Shown are the survival curves ($n = 10$ each). **$P < 0.01$, log-rank test. **d** RUNX1-ETO transduced human CB CD34$^+$ cells were transduced with vector/MT/KR (coexpressing GFP). Shown are the changes of GFP$^+$ (vector/MT/KR transduced) cell frequency. **e**, **f** Cell cycle status and apoptosis of the cells described in (**d**) on day 11. **e** The frequency of S/G2/M phase in GFP$^+$ cells was normalized to that in GFP$^-$ cells ($n = 6$). **f** Shown are the frequency of Annexin V$^+$ cells ($n = 4$). Data are shown as the mean ± s.e.m. **$P < 0.01$, ****$P < 0.0001$, one-way ANOVA with Tukey's multiple comparisons test. **g**, **h** RUNX1-ETO-expressing CB cells were transduced with vector/MT/KR (coexpressing GFP) in combination with vector/BAP1 (coexpressing NGFR) (**g**), with vector/MT (coexpressing GFP) in combination with vector/BAP/BAP1-C91S (CS) (coexpressing NGFR) (**h**). Shown are the changes of GFP/NGFR double-positive (DP) cell frequency. **i** Cell lysates extracted from RUNX1-ETO-expressing CB cells transduced with vector/MT/KR in combination with vector/BAP1 were subjected to immunoblotting

leukemia progression with higher efficiencies than did ASXL1-MT alone. In contrast, ASXL1-MT-K351R showed no effect, indicating that BAP1-induced monoubiquitination of ASXL1-MT plays an important role in enhancing leukaemogenicity (Fig. 5c).

Next, we assessed the role of ASXL1-MT in RUNX1-ETO leukemia using a human cell-based assay. CB cells expressing RUNX1-ETO can grow over 6 months in culture retaining primitive CD34$^+$ cells, and recapitulate many features of human RUNX1-ETO leukemia[32–34]. We transduced a control vector, ASXL1-MT, or ASXL1-MT-K531R (coexpressing GFP) into RUNX1-ETO-expressing CB cells and monitored changes of GFP frequency in culture. Again, ASXL1-MT, but not ASXL1-

MT-K351R, substantially promoted the growth of RUNX1-ETO cells (Fig. 5d).

To determine how ASXL1-MT enhances cell growth, we performed cell cycle and apoptosis analyses. ASXL1-MT consistently increased the proportion of S/G2/M phase cells and decreased the Annexin V$^+$ fraction in human RUNX1-ETO cells (Fig. 5e, f), indicating that ASXL1-MT promotes cell growth through increased cell cycle progression and attenuated apoptosis. Expression of ASXL1-MT resulted in reduction of primitive CD34$^+$ cells and increased expression of CD11b, suggesting that ASXL1-MT does not increase stem cell self-renewal in RUNX1-ETO cells (Supplementary Fig. 5f). To examine the cooperation

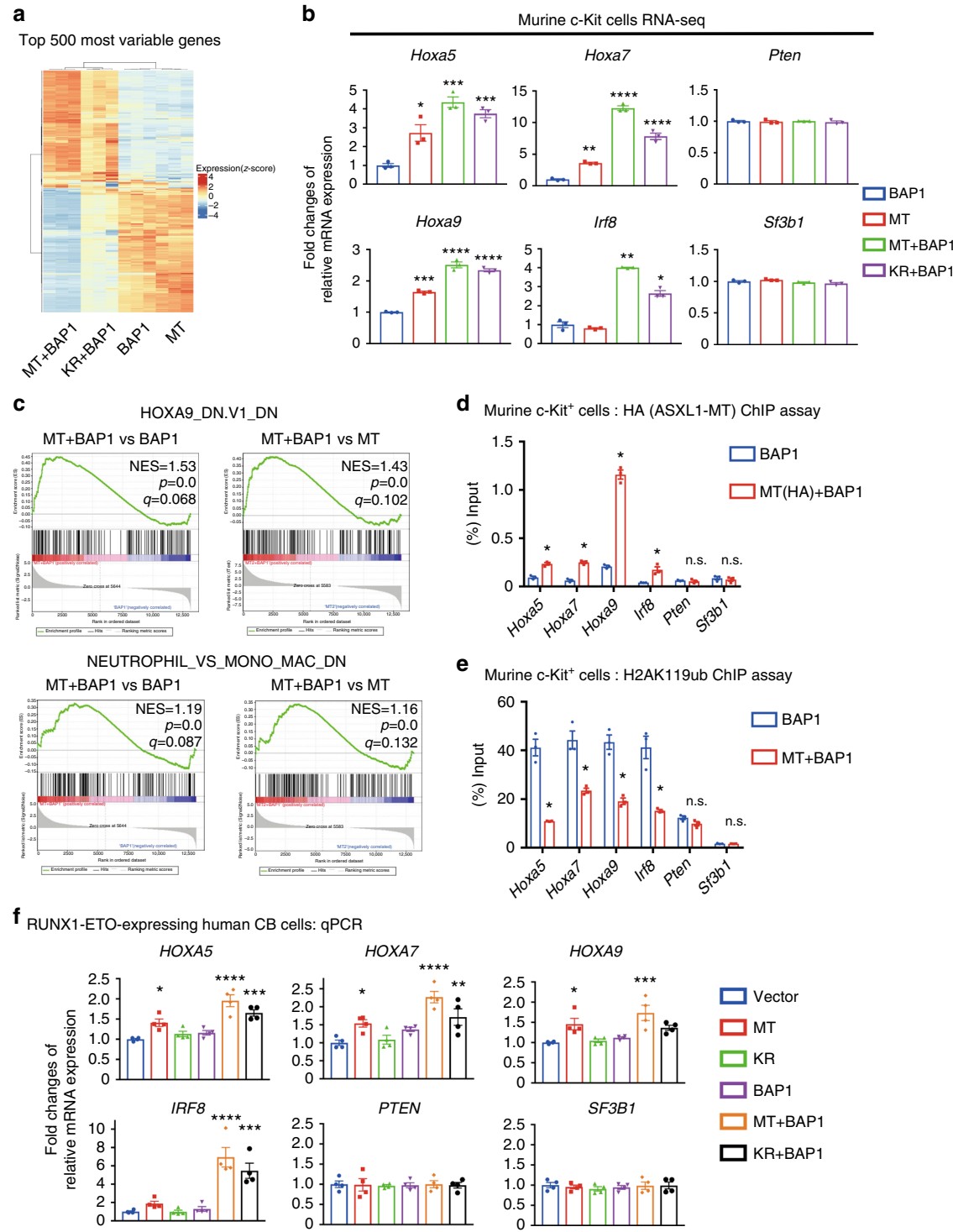

between ASXL1-MT and BAP1 in human RUNX1-ETO cells, we transduced ASXL1-MT or ASXL1-MT-K351R (coexpressing GFP) together with BAP1 or BAP1-C91S (coexpressing NGFR), and monitored changes of the frequency of the GFP+NGFR+ double-positive fraction in culture. Consistent with the results from the mouse model, coexpression of ASXL1-MT and BAP1 showed the strongest growth-promoting effect in RUNX1-ETO cells. In contrast, expression of BAP1-C91S inhibited the growth of RUNX1-ETO cells, suggesting that this catalytically inactive mutant may impair the function of endogenous BAP1 (Fig. 5g, h). We further assessed the cooperative effect of ASXL1-MT and BAP1 on histone modifications in RUNX1-ETO cells (Fig. 5i). In line with our results using murine 32Dcl3 cells, ASXL1-MT was associated with global reduction of H2AK119ub, which was dramatically enhanced by the coexpression of BAP1. Again, ASXL1-MT-K351R showed weaker activity compared with ASXL1-MT.

These data collectively suggest that BAP1-induced monoubiquitination of ASXL1-MT contributes to the increased myeloid proliferation of HSPCs and RUNX1-ETO leukemia cells.

**ASXL1-MT/BAP1 induces upregulation of *HOXA* genes and *IRF8*.** To identify target genes of ASXL1-MT and BAP1, we performed RNA-Seq and ChIP-qPCR analyses using murine c-Kit+ cells transduced with combinations of ASXL1-MT, ASXL1-MT-K351R, and BAP1. Cells coexpressing ASXL1-MT and BAP1 showed increased expression of posterior *Hoxa* cluster genes compared with those expressing ASXL1-MT or BAP1 alone (Fig. 6a, b and Supplementary Data 5, Supplementary Fig. 6a). Furthermore, gene set enrichment analysis (GSEA)[35, 36] showed that many Hoxa9 target genes were upregulated in cells coexpressing ASXL1-MT and BAP1 (Fig. 6c). The upregulation of Hox-target genes was weaker in cells coexpressing ASXL1-MT-K351R and BAP1 (Supplementary Fig. 6b). ChIP-qPCR assay demonstrated binding of ASXL1-MT around promoter loci for *Hoxa5*, *Hoxa7*, and *Hoxa9*, and a significant decrease in H2AK119ub at the same regions. Such epigenetic changes were not observed around promotor loci for *Pten* and *Sf3b1* that were not upregulated by ASXL1-MT and BAP1 (Fig. 6b, d, e).

In addition to posterior *Hoxa* genes, we found that coexpression of ASXL1-MT and BAP1 in c-Kit+ cells led to upregulation of *Irf8*, a master regulator of monocyte differentiation, at both mRNA and protein levels[37] (Fig. 6b and Supplementary Fig. 6a). Coexpression of ASXL1-MT-K351R and BAP1 also increased posterior *Hoxa* gene and *Irf8* expression, but to a lesser extent (Fig. 6b). Downstream target genes of Irf8 (*Klf4* and *Irf5*) were also upregulated in cells expressing both ASXL1-MT and

BAP1[38, 39] (Supplementary Fig. 6c). GSEA revealed that genes upregulated in monocytes/macrophages (compared to neutrophils) were enriched in cells coexpressing ASXL1-MT and BAP1 (Fig. 6c). Furthermore, ASXL1-MT bound to the promoter region of *Irf8*, and H2AK119ub at this region was significantly decreased (Fig. 6d, e). Thus, *Irf8* is also a downstream target of the ASXL1-MT/BAP1 complex, which is likely to underlie the enhanced monocytic differentiation of HSPCs coexpressing ASXL1-MT and BAP1 (Fig. 4b, c). We also examined whether ASXL1-MT and BAP1 upregulated posterior *HOXA* genes and *IRF8* in human RUNX1-ETO cells by qRT-PCR. In line with our expectations, ASXL1-MT upregulated posterior *HOXA* genes and *IRF8*, and coexpression of BAP1 further enhanced this effect. Consistent with earlier results, ASXL1-MT-K351R showed weaker upregulation of these genes (Fig. 6f). Conversely, expression of BAP1-C91S downregulated the expression of posterior *HOXA* genes and *IRF8* in RUNX1-ETO cells (Supplementary Fig. 6d). These data indicate that the ASXL1-MT/BAP1 complex induces upregulation of posterior *HOXA* genes and *IRF8* via removal of H2AK119ub.

**Depletion of BAP1 inhibits ASXL1-MT-induced leukaemogenesis.** To assess the role of endogenous BAP1 in myeloid differentiation, we depleted Bap1 in 32Dcl3 cells using the CRISPR/Cas9 system. 32Dcl3 cells were transduced with Cas9 and two independent guide RNAs (gRNAs) targeting Bap1. Both gRNAs induced nearly complete depletion of Bap1 in 32Dcl3 cells (Fig. 7a). We transduced ASXL1-MT into these control and Bap1-depleted cells, and performed granulocytic differentiation assay. In agreement with our earlier results, expression of ASXL1-MT impaired the G-CSF-induced differentiation of vector-transduced 32Dcl3 cells, as indicated by decreased CD11b expression and altered morphology (Supplementary Fig. 4a). This ASXL1-MT-mediated differentiation block was abrogated in Bap1-depleted cells (Fig. 7b, c). Moreover, BAP1 depletion also abrogated the global reduction of H2AK119ub induced by ASXL1-MT (Fig. 7d). These data suggest that physiological levels of Bap1 expression is required for the inhibition of granulocytic differentiation and reduction of H2AK119ub induced by ASXL1-MT in 32Dcl3 cells.

We next examined whether Bap1 is indispensable for ASXL1-MT-induced leukaemogenesis using murine bone marrow cells transformed by combined expression of SETBP1-D868N and ASXL1-MT (cSAM cells: cells with combined expression of SETBP1 and ASXL1 Mutations). SETBP1-D868N is an oncogenic mutation of SETBP1, which collaborates with ASXL1-MT in the induction of myeloid transformation[19]. We transduced Cas9 and Bap1-targeting gRNAs into cSAM cells, and cultured the cells in

**Fig. 6** ASXL1-MT/BAP1 complex induces upregulation of posterior *HOXA* genes and *IRF8* through removal of H2AK119ub. **a** Murine bone marrow c-Kit+ cells were transduced with ASXL1-MT (MT) or ASXL1-MT-K351R (KR) (coexpressing blasticidin-resistant gene) together with vector or BAP1 (coexpressing puromycin-resistant gene), and were cultured in M3234 containing 20 ng/ml SCF, 10 ng/ml IL-3, and 10 ng/ml IL-6. After the selection with blasticidin and puromycin for 3 days, colony-forming cells were collected to extract RNA for RNA-seq analysis. A heatmap of the top 500 differentially expressed genes is shown. **b** Relative expression of *Hoxa5*, *Hoxa7*, *Hoxa9*, *Irf8*, *Pten*, and *Sf3b1* in the indicated cells. Expression of each gene in cells expressing BAP1 alone was set as 1. Data are shown as mean ± s.e.m. of three biologically independent experiments. *$P < 0.05$, **$P < 0.01$, ****$P < 0.0001$, one-way ANOVA with Tukey's multiple comparisons test. **c** Gene set enrichment analysis (GSEA) revealed that Hoxa9 target genes (upper) and genes related to monocyte/macrophage (lower) were highly expressed in cells coexpressing ASXL1-MT and BAP1 compared with those in cells expressing ASXL1-MT or BAP1 alone. **d, e** Murine bone marrow c-Kit+ cells were transduced with BAP1 together with vector or HA-tagged ASXL1-MT, and were cultured in semisolid medium. Genomic DNA fragments from these cells were immunoprecipitated with anti-HA (**d**) and anti-H2AK119ub (**e**) antibodies. Enrichments of H2AK119ub and ASXL1-MT (HA) at *Hoxa5*, *Hoxa7*, *Hoxa9*, *Irf8*, *Pten*, and *Sf3b1* promotor loci were measured by qPCR. Data are shown as mean ± s.e.m. of three biological independent experiments. *$P < 0.05$, Student's *t*-test. **f** RUNX1-ETO-expressing CB cells were transduced with vector/MT/KR (coexpressing GFP) in combination with vector/BAP1. GFP/NGFR double-positive cells were sorted 48 h after transduction. Relative mRNA levels of *HOXA5*, *HOXA7*, *HOXA9*, *IRF8*, *PTEN*, and *SF3B1* were analyzed by qRT-PCR. Results were normalized to *GAPDH*, with the relative mRNA level in vector-transduced cells set at 1. Data are shown as mean ± s.e.m. of four biological independent experiments. *$P < 0.05$, **$P < 0.01$, ***$P < 0.001$, ****$P < 0.0001$, one-way ANOVA with Dunnett's multiple comparisons test

semisolid medium or directly transplanted them into sublethally irradiated recipient mice (Fig. 7e). Bap1 depletion resulted in a profound reduction in the numbers and sizes of cSAM cell colonies (Fig. 7f and Supplementary Fig. 7a), and also delayed the development of leukemia in mice, suggesting an important role for Bap1 in ASXL1-MT-induced leukemogenesis (Fig. 7g). Ectopic expression of human BAP1 rescued the reduced colony formation of cSAM cells induced by Bap1 depletion (Fig. 7f and Supplementary Fig. 7b, c). In addition, Bap1 depletion in cSAM cells led to a significant decrease in posterior *Hoxa* gene expression, which probably underlies the reduced leukemogenicity of cSAM cells with Bap1 depletion (Fig. 7h). These results suggest that the impairment of myeloid differentiation,

upregulation of posterior *Hoxa* genes, and promotion of myeloid leukaemogenesis by ASXL1-MT are dependent on endogenous Bap1.

***Hoxa* genes and *Irf8* are critical targets of ASXL1-MT/BAP1**. To assess the role of *Hoxa* genes in ASXL1-MT/BAP1-induced leukemogenesis, we depleted Hoxa5, Hoxa7, or Hoxa9 in cSAM cells using CRISPR/Cas9 system and assessed their colony-forming ability. Depletion of each of these individual *Hoxa* genes partially decreased the colony-forming activity of cSAM cells (Fig. 8a and Supplementary Fig. 8a). In contrast, depletion of Irf8 in cSAM cells showed no effect on their colony formation. We

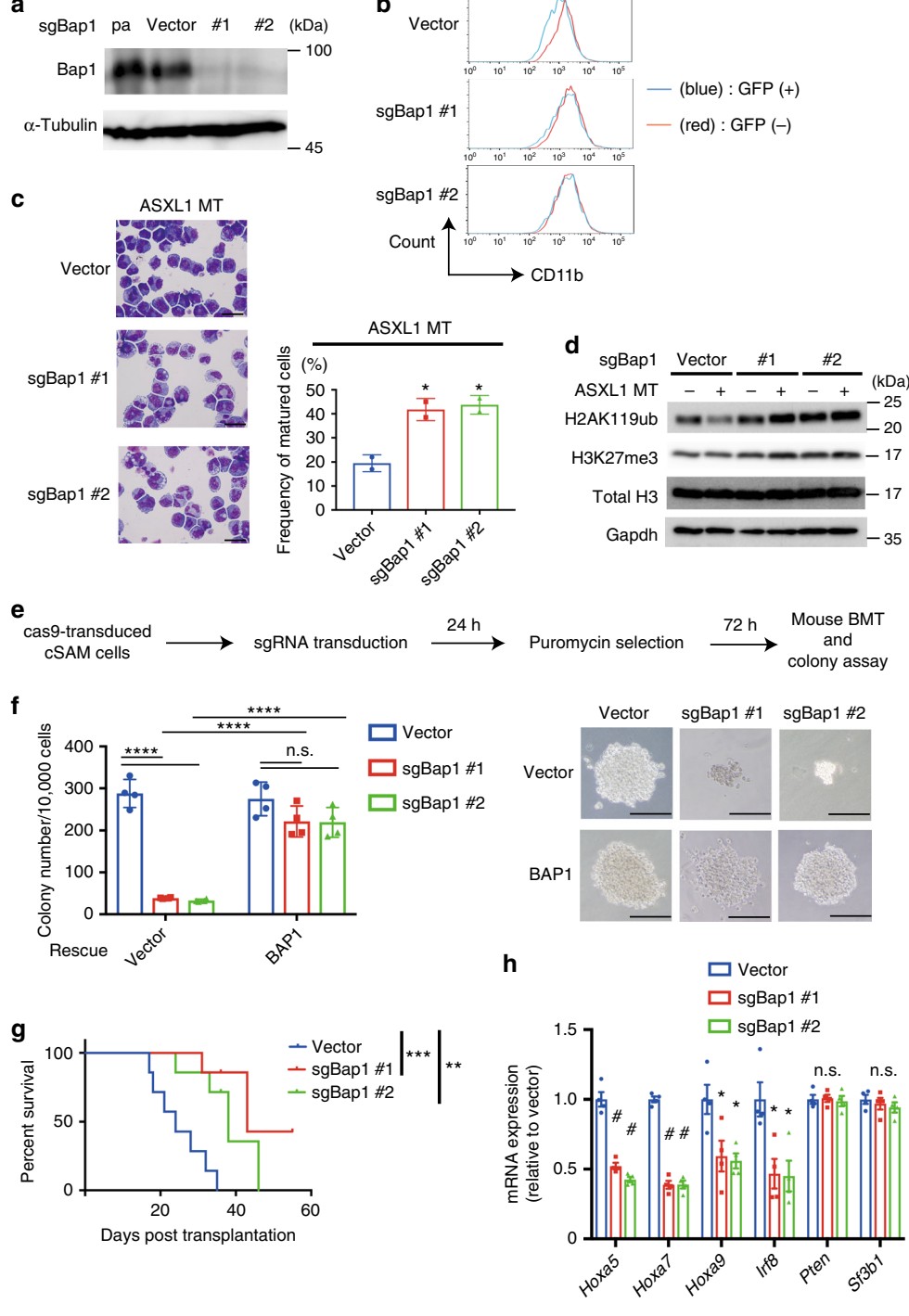

then examined if ectopic expression of *Hox* genes in cSAM cells can reverse the inhibitory effect of BAP1 depletion on colony formation. As shown in Fig. 8c, d, forced expression of HOXA7 and HOXA9 rescued the reduced colony-forming activity of Bap1-depleted cSAM cells.

We then transduced RUNX1-ETO9a, ASXL1-MT, and BAP1 into bone marrow cells derived from Rosa26-LSL-Cas9 knockin mouse[40] to establish RUNX1-ETO leukemia cells expressing ASXL1-MT, BAP1, and Cas9. Similar to the previous results, depletion of Hoxa5, Hoxa7, or Hoxa9 resulted in the partial reduction of their colony-forming capacity (Fig. 8b and Supplementary Fig. 8b). Taken together, these results suggest that posterior *Hoxa* genes contribute to leukemogenesis induced by ASXL1-MT and BAP1.

Next, we assessed the role of Irf8 on ASXL1-MT/BAP1-induced monopoiesis. We transduced ASXL1-MT, BAP1, and *Irf8*-targeting sgRNA into c-kit[+] bone marrow cells from Rosa26-LSL-Cas9 knockin mouse, and cultured these cells with cytokines to promote myeloid differentiation. Irf8 depletion partially reduced the frequency of monocytes, whereas restored that of granulocytes in ASXL1-MT/BAP1-expressing cells (Fig. 8e, f and Supplementary Fig. 8c, d). Thus, these data indicate that upregulation of Irf8 underlies the enhanced monopoiesis induced by ASXL1-MT and BAP1.

**BAP1 has a growth-promoting role in myeloid leukemia cells.** Because the dysregulation of HOX genes is commonly observed across a range of disparate AML subtypes[41], we speculated that BAP1 might also support the growth of other myeloid leukemia cells. To test this possibility, we assessed the role of Bap1 in MLL-fusion leukemia, a well-known AML subtype characterized by *Hox* gene dysregulation[42]. Mouse bone marrow c-Kit[+] cells were transduced with an oncogenic MLL-fusion gene MLL-AF9, and serially replated up to five times to generate murine MLL-AF9 leukemia cells. We then transduced Cas9 and Bap1-targeting gRNAs (sgBap1 #1 and #2) into the MLL-AF9 cells, and examined the effect of BAP1 depletion in MLL-AF9 leukemia cells. Similar to our results using cSAM cells, Bap1 depletion profoundly inhibited the colonogenicity and leukaemogenicity of MLL-AF9 leukemia cells, and this effect was rescued by expression of exogeneous BAP1 (Fig. 9a, b and Supplementary Fig. 9a). Furthermore, Bap1 depletion resulted in reduction of posterior *Hoxa* gene expression (Fig. 9c), demonstrating an obligate role for Bap1 in MLL-AF9-driven leukaemogenesis.

Finally, we assessed the role of BAP1 in several human myeloid leukemia cell lines. Kasumi-1 is characterized by an ASXL1

mutation (G646WfsX12) and RUNX1-ETO fusion. MEG-01 and TS9;22 are other myeloid leukemia cell lines with ASXL1 mutations (G646WfsX12 in MEG-01 and R693X in TS9;22). MOLM-13 and THP-1 are cell lines harboring MLL-AF9. We transduced Cas9 and gRNAs targeting human BAP1 (sgBAP1 #3 and #4) into these cells. BAP1 depletion resulted in substantial growth inhibition in all of these myeloid leukemia cell lines. We also confirmed downregulation of posterior *HOXA* genes and *IRF8* expression in BAP1-depleted MEG-01, TS9;22, MOLM-13, and THP-1 cells (Supplementary Fig. 9b). In contrast, the growth of a T cell leukemia cell line (Jurkat) and a B cell lymphoma cell line (Raji) were unaffected by BAP1 depletion (Fig. 9d). Thus, BAP1 has a growth-promoting effect in a wide range of myeloid leukemia cells, including AMLs with ASXL1 mutations or MLL-fusions.

## Discussion

*ASXL1* mutations are frequently identified in a variety of myeloid neoplasms and are associated with poor prognosis, making the development of molecular therapies targeting these mutations an important goal. Our data indicate an obligate role for BAP1 in mutant ASXL1-induced myeloid leukaemogenesis, and suggest this enzyme as a promising therapeutic target. Research to date has primarily focused on the tumor-suppressing role of BAP1. Clinical data have shown that *BAP1* gene is frequently mutated in various tumors, including mesothelioma and melanoma[43–45]. Experimentally, it has been shown that *Bap1* deletion in adult mice causes MDS-like diseases[46]. Although these studies have clearly shown a tumor suppressor role for BAP1, several lines of evidence also raise the possibility that BAP1 activity drives the growth of myeloid leukemia cells with *ASXL1* mutations. First, *BAP1* is rarely mutated in patients with myeloid neoplasms, and *BAP1* and *ASXL1* genes are mutated in distinct cancer types[47]. Second, BAP1 and its *Drosophila* homolog Calypso have been shown to maintain posterior *HOXA* genes and it has also been shown that *ASXL1* deletion/mutations lead to increased expression of posterior *HOXA* genes[16, 17]. Third, one recent study showed that BAP1 loss results in transformation, independent of ASXL1 activity[48]. In addition to these reports implicating BAP1 in myeloid transformation, the present results identify the underlying molecular mechanisms by which ASXL1-MT/BAP1 complex induce myeloid transformation; BAP1 plays a necessary role in maintaining aberrant posterior *HOXA* expression in *ASXL1*-mutant leukemia and in sustaining their leukemic proliferation. Maintenance of the leukemic phenotype in MLL-fusion leukemia, a well-known AML subtype with posterior *HOXA*

**Fig. 7** Depletion of BAP1 abrogates ASXL1-MT-induced differentiation block and inhibits leukaemogenesis. **a** 32Dcl3 cells were first transduced with Cas9, and were then transduced with a vector control or two independent sgRNAs targeting Bap1. Cell lysates were extracted from these cells and parent 32Dcl3 cells (pa), followed by immunoblotting analysis. **b** 32Dcl3 cells described in (**a**) were transduced with ASXL1-MT (coexpressing GFP), and were cultured with 50 ng/ml G-CSF. CD11b expression in untransduced (GFP−, red line) and transduced (GFP+, blue line) cells was assessed on day 6. **c** Morphology of the cells described in (**b**) was assessed by Wright–Giemsa staining. Original magnification, ×1000; Scale bars: 20 μm (left). Two independent experiments were performed, and proportions of segmented cells are shown as mean ± s.e.m. (right). *P < 0.05, one-way ANOVA with Dunnett's multiple comparisons test. **d** Cell lysates extracted from 32Dcl3 cells described in (**b**) were subjected to immunoblotting. **e** Experimental scheme used in (**f–h**). Primary leukemia cells transformed by ASXL1-MT and SETBP1-D868N (cSAM cells) were transduced with Cas9 and Bap1-targeting sgRNAs, and were cultured in semisolid medium or transplanted into recipient mice. **f** Bap1-depleted cSAM cells produced significantly reduced numbers of colonies compared with control cSAM cells. Expression of human BAP1 reversed the reduced colony formation of Bap1-depleted cSAM cells. Colony numbers were counted on day 7 (*n* = 4). Data are shown as mean ± s.e.m. ****P < 0.0001, two-way ANOVA with Tukey's multiple comparison test (left). Representative photos of colonies are also shown (right). Scale bars: 200 μm. **g** Survival curves of recipient mice transplanted with vector- or sgBAP1-transduced cSAM cells (*n* = 7 each). **P = 0.0031, ***P = 0.0008, log-rank test. **h** Relative expression of *Hoxa5*, *Hoxa7*, *Hoxa9*, *Irf8*, *Pten*, and *Sf3b1* were assessed in vector- or sgBAP1-transduced cSAM cells using qRT-PCR. Results were normalized to *Gapdh*, with the relative mRNA level in vector-transduced cells set at 1. Data are shown as mean ± s. e.m. of four biological independent experiments. *P < 0.05, #P < 0.01, one-way ANOVA with Dunnett's multiple comparisons test

dysregulation[49, 50], also depends on the BAP1 activity. Inhibition of BAP1 activity, rather than enhancing it, may thus represent a promising therapeutic strategy for myeloid neoplasms involving posterior *HOXA* dysregulation.

The ASXL1-MT/BAP1 complex enhances transient myeloid proliferation, but does not increase the long-term proliferation

potential of murine and human HSPCs. Thus, unlike other epigenetic genes related to clonal haematopoiesis (e.g., DNMT3A and TET2) whose disruption leads to increased HSPC function[51, 52], it appears that *ASXL1* mutations do not confer the capacity for self-renewal to HSPCs. The ASXL1-MT/BAP1 complex is not a simple growth-accelerator, but rather inhibits

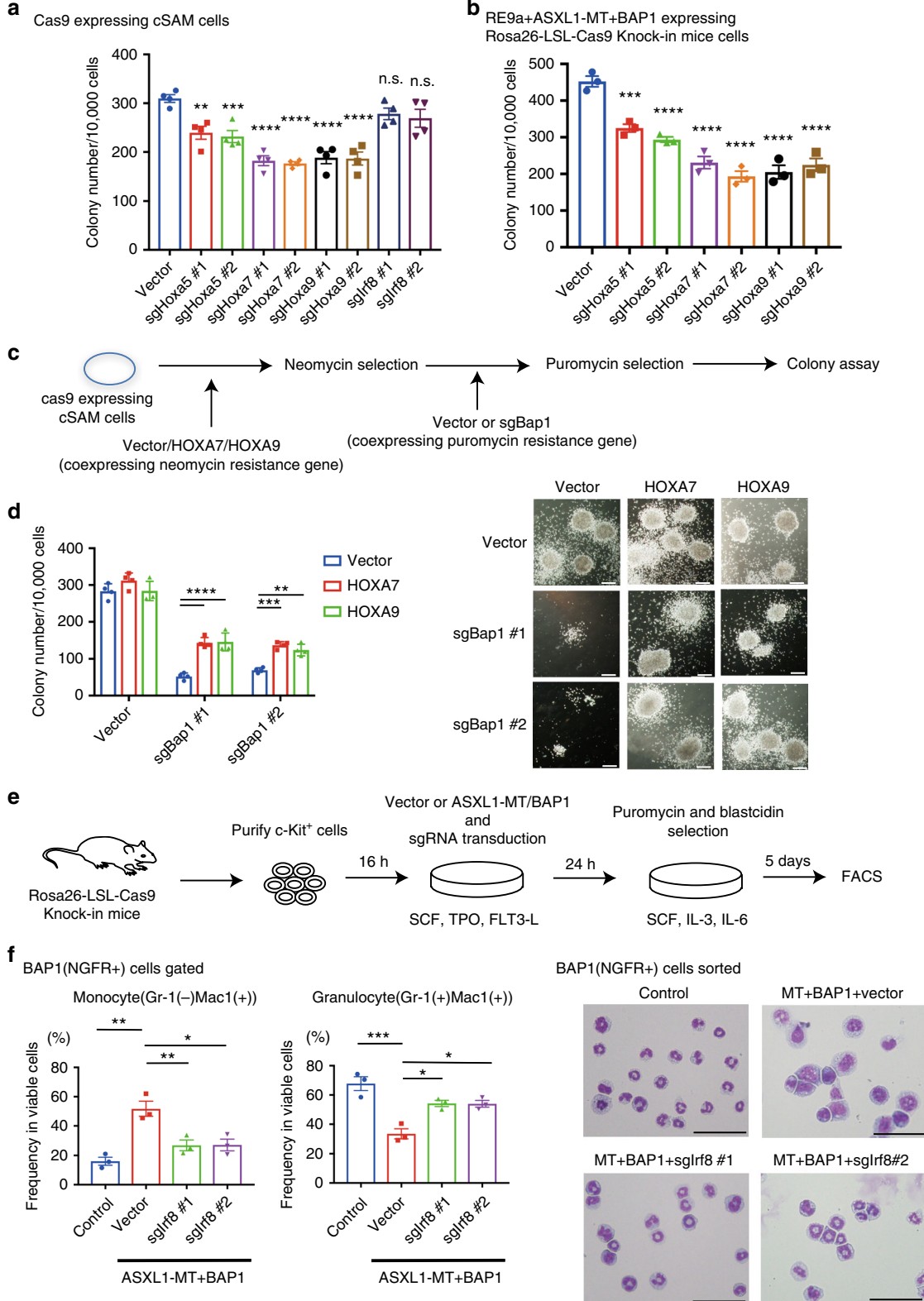

terminal differentiation of HSPCs toward granulocytes, macrophages, mast cells, and erythrocytes. Interestingly, ASXL1-MT/BAP1 promotes monocyte differentiation, while it inhibits terminal differentiation to macrophage, which may account for the frequent (40–50%) detection of ASXL1 mutations in CMML patients. We also identified *IRF8*, a master regulator of monocyte differentiation, as a novel target gene of the ASXL1-MT/BAP1 complex. Consistent with this finding, one previous study showed substantial downregulation of *IRF8* in *Bap1*-deleted murine HSPCs[46].

We have shown here that BAP1 recruits ubiquitin ligases to induce both mono- and polyubiquitination of ASXL1-MT. Importantly, BAP1 concurrently removes polyubiquitinated chains from ASXL1-MT, leaving only the monoubiquitinated form. Monoubiquitinated ASXL1-MT in turn enhances the catalytic activity of BAP1, creating a positive feedback loop that establishes the hyperactive ASXL1-MT/BAP1 complex. We additionally identified UBE2O as a candidate ligase that promotes polyubiquitination and degradation of ASXL1-MT. Given that UBE2O has been shown to inhibit BAP1 function[23], activation of UBE2O may represent an alternative approach to suppress the hyperactivity of the ASXL-MT/BAP1 complex. However, UBE2O is dispensable for BAP1-induced monoubiquitination of ASXL1-MT at lysine 351. We also examined the roles of additional 19 candidate ubiquitination-related proteins in 293T cells that coexpress ASXL1-MT, BAP1, and Cas9, but depletion of any individual ubiquitination-related protein did not reduce the monoubiquitination of ASXL1-MT (Supplementary Fig. 10 and Supplementary Table 1). Therefore, it is likely that BAP1-induced monoubiquitination of ASXL1-MT is mediated by multiple E3 ligases with redundant functions or mediated by some unknown E3 ligase(s). Mechanisms underlying the enhancement of the catalytic activity of ASXL1-MT/BAP1 complex warrant further investigation. A previous report showed that the deubiquitinase adapter (DEUBAD) domain of ASXL1 increases BAP1's affinity for ubiquitin on H2A[15], and we showed that monoubiquitinated ASXL1-MT enhances nuclear retention of BAP1. These findings suggest that ASXL1-MT promotes the recruitment of BAP1 to chromatin, especially toward the regions with H2AK119ub. However, this concept needs to be experimentally demonstrated in future research.

In summary, our study demonstrates the critical role of BAP1 in ASXL1-MT-induced aberrant myeloid differentiation, myeloid leukaemogenesis, and upregulation of *HOXA* genes and *IRF8*. Targeting enzymatic activity of BAP1 can be a promising therapeutic strategy for myeloid neoplasms with ASXL1 mutations, and potentially for a broad range of myeloid neoplasms with HOX dysregulation.

## Methods

**Murine myeloid colony assay and transplantation assay**. All animal studies were approved by the Animal Care Committee of the Institute of Medical Science at the University of Tokyo (approval number: K15-32, PA13-19 and A17-75), and were conducted in accordance with the Regulation on Animal Experimentation at University of Tokyo based on International Guiding Principles for Biomedical Research Involving Animals. Mouse bone marrow c-Kit[+] cells derived from C57BL/6 (Ly5.2) 8- to 12-week-old male mice (Charles River Laboratories Japan, Yokohama, Japan) transduced with the combinations of ASXL1-MT, ASXL1-MT-K351R, and BAP1 were plated in M3234 (STEMCELL Technologies) methylcellulose containing 20 ng/ml SCF, 10 ng/ml IL-3, and IL-6. A total of $1 \times 10^4$ cells were plated for each round of plating. Colony counting and replating were performed every 7 days. To generate RUNX1-ETO9a leukemia in vivo, E14.5 c-kit[+] fetal liver cells derived from C57BL/6 (Ly5.2) mice were transduced with RUNX1-ETO9a together with the combinations of ASXL1-MT, ASXL1-MT-K351R and BAP1, and were transplanted into sublethally irradiated 8- to 12-week-old female C57B/6J (Ly5.1) mice (Sankyo Labo Service Corporation, Tokyo, Japan)[53]. cSAM cells and MLL-AF9 cells were generated by transducing ASXL1-MT and SETBP1-D868N or MLL-AF9 into murine bone marrow progenitors, respectively[19, 54]. In some experiments, these cells were transduced with Cas9 and Bap1-targeting sgRNAs, and were transplanted into sublethally irradiated 8- to 12-week-old female C57B/6J (Ly5.1) mice. We also used c-Kit[+] cells derived from 8- to 12-week-old Rosa26-LSL-Cas9 knockin male mice[40] (Jackson Laboratory) to establish RUNX1-ETO9a leukemia cells that coexpress ASXL1-MT, BAP1, and Cas9. Randomization and blinding were not performed in this study.

**Cell culture**. HEK293T (CRL-11268, ATCC, Manassas, VA, USA) and Hela (CCL-2, ATCC) were cultured in DMEM containing 10% fetal bovine serum (FBS). 32Dcl3 (CRL-11346, ATCC) were cultured in RPMI-1640 medium supplemented with 10% FBS and 1 ng/ml IL-3. K562 (CCL-243, ATCC), Kasumi-1 (CRL-2724, ATCC), MEG-01 (CRL-2021, ATCC), TS9;22[55], MOLM-13 (ACC-554, DSMZ, Braunschweig, Germany), THP-1 (TIB-202, ATCC), Jurkat (TIB-152, ATCC), and Raji (CCL-86, ATCC) were cultured in RPMI-1640 medium supplemented with 10% FBS. TF-1 cells were cultured in RPMI-1640 medium supplemented with 10% FBS and 2 ng/ml GM-CSF. For erythroid differentiation assay, TF-1 cells were washed with PBS three times and were incubated with RPMI-1640 supplemented with 10% FBS and 2 U/ml erythropoietin (EPO)[56]. All cell lines used in this study were authenticated by short-tandem repeat analyses and tested for mycoplasma contamination in our laboratory. Mouse bone marrow c-Kit[+] cells were separated using CD117 MicroBead Kit (Miltenyi Biotec) and were cultured in Iscove's modified Dulbecco's media (IMDM) containing 20% FBS and 50 ng/ml murine SCF, TPO, and FLT-3L for 16 h, and then were induced differentiation. For the mast cell maturation assay, we incubated cells with IMDM containing 20% FBS, 50 ng/ml SCF, 10 ng/ml IL-3 and IL-6[21]. For the monocyte/macrophage differentiation assay, cells were incubated with DMEM containing 10% FBS and 100 ng/ml macrophage colony-stimulating factor (M-CSF)[25]. Human CB cells were obtained from Riken BRC or the Japanese Red Cross Kanto-Koshinetsu Cord Blood Bank (Tokyo, Japan) following an institutional review board-approved protocol.

**Fig. 8** *Hoxa* genes and *Irf8* contribute to leukaemogenesis and monopoiesis induced by ASXL1-MT and BAP1. **a** Cas9-expressing cSAM cells were transduced with vector or two independent sgRNAs targeting mouse Hoxa5, Hoxa7, Hoxa9, or Irf8. Colony-forming activity was assessed using these cells. Shown are the colony counts per $10^4$ plated cells from quadricate plates. Hoxa5, Hoxa7, or Hoxa9 depletion reduced the colony-forming activity of cSAM cells. One-way ANOVA with Dunnett's multiple comparisons test. **b** Mouse bone marrow cells expressing RUNX1-ETO9a, ASXL1-MT, BAP1, and Cas9 were transduced with vector or two independent sgRNAs targeting mouse Hoxa5, Hoxa7, or Hoxa9. Colony-forming activity was assessed using these cells. Shown are the colony counts per $10^4$ plated cells from triplicate plates. Hoxa5, Hoxa7, or Hoxa9 depletion reduced the colony-forming activity of RUNX1-ETO9a, ASXL1-MT, and BAP1-expressing cells. One-way ANOVA with Dunnett's multiple comparisons test. **c** Schematic presentation of experimental procedures for experiments shown in Fig. 8d. Cas9-expressing cSAM cells were transduced together with vector, HOXA7, or HOXA9 (coexpressing neomycin resistance gene). After G418 (1 mg/ml) selection for 7 days, cells were transduced with vector or two independent sgRNAs targeting mouse Bap1 (sgBap1 #1, #2). **d** Ectopic expression of HOXA7 or HOXA9 partially reversed the reduced colony-forming activity of Bap1-depleted cSAM cells. Colony numbers were counted on day 7 (n = 4). Two-way ANOVA with Tukey's multiple comparison test. **e** Schematic presentation of experimental procedures for experiments shown in Fig. 8f. c-Kit[+] bone marrow cells were purified from Rosa-LSL-Cas9 mice and were then retrovirally transduced with control vectors or ASXL1-MT (coexpressing blastcidin resistance gene) and BAP1 (coexpressing NGFR)-expressing vectors, and were lentivirally transduced with control vector or sgRNAs (coexpressing puromycin resistance gene) targeting mouse Irf8. After puromycin and blastcidin selection, cells were cultured with cytokines to induce differentiation toward both granulocytes and monocytes. **f** Granulocyte (Gr-1[+]CD11b[+]) and monocyte (Gr-1[−]CD11b[+]) differentiation was assessed by FACS analyses (left) and Wright–Giemsa staining (right, scale bars: 50 μm) after 5 days of culture (n = 3). One-way ANOVA with Tukey's multiple comparisons test. Data are shown as mean ± s.e.m. *P < 0.05, **P < 0.01, ***P < 0.001, ****P < 0.0001, n.s. not significant

Informed consent was obtained in accordance with the Declaration of Helsinki. CD34[+] cells were separated using CD34 MicroBead Kit (Miltenyi Biotec). For myeloid differentiation assay, cells were cultured in StemSpan (STEMCELL Technologies, Bancouber, BC, Canada) containing 10 ng/ml human SCF, mega-karyocyte growth and development factor (TPO), FLT3 ligand (FLT-3L), IL-3, and IL-6 (R&D Systems, Minneapolis, MN)[33, 34]. For erythroid differentiation assay, cells were cultured in StemSpan containing 100 ng/ml SCF, 2 U/ml erythropoietin (EPO), insulin, transferrin, selenium solution (ITS-G) (Thermo Fisher scientific), and insulin growth factor-1 (IGF-1) (SIGMA, #I3769), and 10 μM dexamethasone (Wako, JAPAN)[57]. In stem cell culture condition, cells were cultured in StemSpan containing 50 ng/ml human SCF, TPO, FLT-3L, and 750 nM stemreginin-1[58].

**Plasmids**. We used ASXL1-WT, ASXL1-MT, ASXL1-G646WfsX12, and ASXL1-MT-K351R in the pMYs-IRES-GFP (pMYs-IG) vector[22] or pcDNA3.1 vector.

We also used 3xFLAG-tagged, and Myc-tagged ASXL1-MT, FLAG-tagged and Myc-tagged ASXL1-MT-K351R, FLAG-tagged ASXL1-Y591X and Myc-tagged ASXL1-WT in the pMYs-IRES-GFP vector and HA-tagged ASXL1-MT in the pMYs-IRES-Blastcidin vector. For BAP1 expression, we used pEF-4xHA-BAP1 WT, pMMP-puro-BAP1 WT, and pMMP-puro-BAP1 C91S [gifts from Dr. Machida[59]], and HA-tagged BAP1 WT/C91S in pMYs-IRES-nerve growth factor receptor (NGFR) vector and pMYs-IRES-tdTomato vector. For UBE2O expression, we used pLV-Myc-UBE2O WT and pLV-Myc-UBE2O C885S [gifts from Dr. Dijke[60]]. For RUNX1-ETO expression, we used HA-tagged RUNX1-ETO or RUNX1-ETO9a in a pMSCV-IRES-Thy1.1 retroviral vector[54]. For MLL-AF9 expression, we used pMSCV-MLL-AF9-pgk-EGFP retroviral vector[54]. For HOXA7 and HOXA9 expression, we used Flag-tagged HOXA7 and HOXA9 in pMSCV-neo retroviral vector[54]. The full-length cDNAs of HOXA7 and HOXA9 were purchased from Promega, and we cloned them into the pMSCV-neo vector (Clontech).

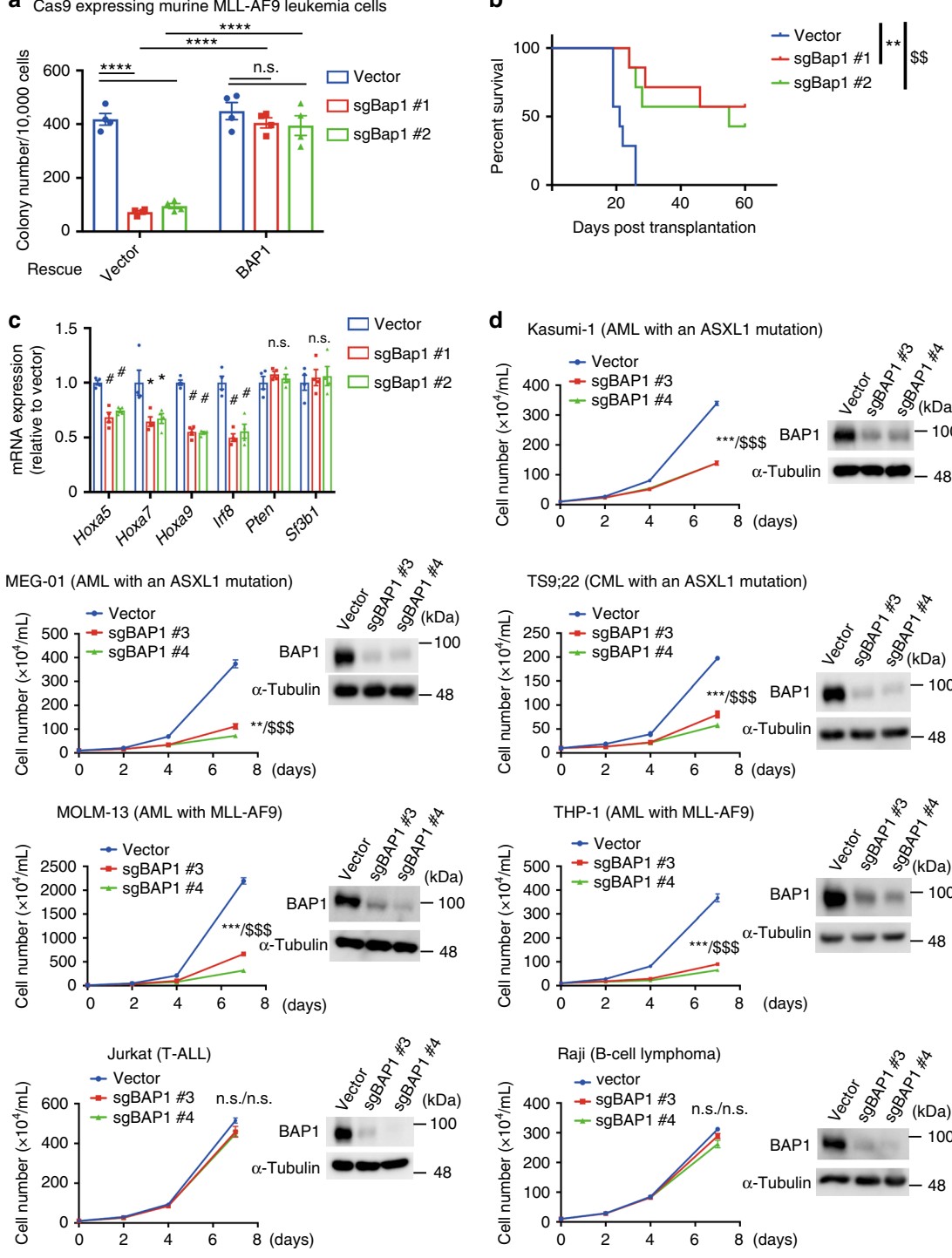

**Viral transduction**. Retroviruses for mouse cells were produced by transient transfection of Plat-E packaging cells with retroviral constructs using the calcium-phosphate method[61]. Retroviruses for human cells and lentiviruses were produced by transient transfection of 293T cells with viral plasmids along with gag-, pol-, and env-expressing plasmids using the calcium-phosphate method[34]. Retrovirus transduction to the cells was performed using Retronectin (Takara Bio Inc., Otsu, Shiga, Japan).

**Gene depletion using the CRISPR/Cas9 system**. To generate short guide RNA (sgRNA) constructs, annnealed oligos were inserted into pLentiguide-puro vector[62], which was obtained from Addgene (#52963). Cas9-expressing vector (lentiCas9-Blast #52962); and lentiviral packaging vectors [(pMD2.G #12259) and (psPAX2 #12260)] were also purchased from Addgene. Lentiviruses were generated by transient transfection of 293T cells with these lentiviral constructs. Cells were infected with the virus for 24 h, and were selected for stable expression of Cas9 using blasticidin (10 μg/ml) and for stable expression of sgRNAs using puromycin (1 μg/ml). The cells were then harvested to examine the depletion of targeted gene by immunoblotting. Sequence of the sgRNAs targeting for mouse Bap1, human BAP1, human UBE2O, mouse Hoxa5, mouse Hoxa7, mouse Hoxa9, and mouse Irf8 from 5′ to 3′ are provided as follows: CCTGATCGTAGGTGTCAAAG (sgBap1#1), AGTGGACAGATAAAGCTCGA (sgBap1#2), GAACCGTCAGACA GTACTAG (sgBAP1#3), TCTACCCCATTGACCATGGT (sgBAP1#4), GCCC GACGTAGAGCGCAAGG (sgUBE2O#1), GACTTCGTGGTAGATAAGCG (sgUBE2O#2), CCCCACTCGAACCCCCTACG (sgUBE2O#3), AATGGCATGGA TCTCAGCGT (sgHoxa5#1), GTCCCTGAATTGTTCGCTCA (sgHoxa5#2), GCCTGCGACAAGGCGGACGA (sgHoxa7#1), CCTGGATGCGCAGTTCAGGT (sgHoxa7#2), CGCGTGCACTGGGTTCCACG (sgHoxa9#1), ACCGGGCCATTA ATAGCGTG (sgHoxa9#2), AGTTTACCGAATTGTCCCCG (sgIrf8#1), and TCGACAGCAGCATGTACCCG (sgIrf8#2).

**Immunofluorescence analysis**. 293T cells transfected with FLAG-tagged vector, ASXL1 (MT or MT-K351R) and HA-tagged BAP1 (WT or C91S) were fixed with 2% paraformaldehyde, permealized with 0.2% Triton-X, blocked with 2% BSA and 5% goat serum, and were then incubated with anti-FLAG rabbit monoclonal antibody (Sigma-Aldrich, catalog #F7425, 1:200) and anti-HA mouse monoclonal antibody (BioLegend, catalog #901513, 1:200), followed by labeling with Alexa Fluor 568-conjugated anti-rabbit (Thermo Fisher, catalog #A11011, 1:1000) and Alexa Fluor 488-conjugated anti-mouse antibody (Thermo Fisher, catalog #A11029, 1:1000). Nuclei were visualized with DAPI (BioLegend catalog #422801, 1 μg/ml). Fluorescent images were analyzed on a confocal microscope (Nikon A1).

**Flow cytometry**. Cells were stained by fluoro-conjugated antibodies for 30 min at 4 °C. After staining, cells were washed with cold PBS for several times, and were resuspended with PBS containing 2% FBS. Cells were analyzed on a FACSCalibur or a FACS Verse, and were sorted with a FACSAria (BD Biosciences, San Jose, CA, USA). Cell cycle analysis (Vybrant DyeCycle Violet stain; Invitrogen) and apoptosis analysis (Annexin V-APC kit; BD Biosciences) were performed according to the manufacturer's recommendations. Analyses were performed using Flowjo software (Tree Star Inc.). The following fluoro-conjugated antibodies were used in this study at indicated dilutions: anti-mouse B220 (eBioscience, catalog #11-0452-82, clone RA3-6B2, 1:200), anti-mouse CD11b (eBioscience, catalog #14-0112-81, clone M1/70, 1:200), anti-mouse CD3 (eBioscience, catalog #11-0036-42, clone 145-2C11, 1:200), anti-mouse CD117 (BioLegend, catalog #105808, clone 2B8, 1:200), anti-mouse F4/80 (BioLegend, catalog #123109, clone BM8, 1:100), anti-mouse Gr-1 (BioLegend, catalog #108411, clone RB6-8C5, 1:400), anti-mouse Fce1Rα (eBioscience, catalog #12-5898-81, clone MAR-1, 1:200), anti-mouse CD115 (BioLegend, catalog #135509, clone AFS98, 1:100), anti-mouse CD11b

(BioLegend, catalog #101215, clone M1/70, 1:200), anti-mouse CD45.2 (BioLegend, catalog #109813, clone 104, 1:200), anti-mouse CD90.1 (Thy-1.1) (BioLegend, catalog #202506, clone OX-7, 1:400), anti-human CD33 (BioLegend, catalog #303403, clone WM53, 1:200), anti-human CD66b (BioLegend, catalog #305115, clone G10F5, 1:200), anti-human CD235ab (BioLegend, catalog #306607, clone HIR2, 1:200), anti-human CD71 (BioLegend, catalog #334105, clone CY1G4, 1:200), anti-human CD34 (BioLegend, catalog #343607, clone 561, 1:200), anti-human CD271 (NGFR) (BioLegend, catalog #345105 and #345017, clone ME20.4, 1:400).

**Immunoprecipitation and western blotting analysis**. 293T cells were transiently transfected with plasmids using polyethylenimine (PEI). Forty-eight hours after transfection, cells were lysed with cell lysis buffer (Cell Signaling Technology, Danvers, MA, USA; #9803). For immunoprecipitation, cell lysates were incubated with anti-FLAG (Sigma-Aldrich, catalog F1804, clone M2, 1:200) or anti-HA (Santa Cruz Biotechnology, catalog sc-57592, clone 12CA5, 1:100) antibody for 30 min at 4 °C. The samples were then incubated with protein-G-Sepharose (Amersham Pharmacia Biotech, Piscataway, NJ, USA) for 30 min at 4 °C. The precipitates were washed stringently three times with lysis buffer (Cell Signaling Technology, Danvers, MA, USA; #9803) containing 1 mM phenylmethanesulfonyl fluoride, subjected to sodium dodecyl sulfate-polyacrylamide gel electrophoresis, and were analyzed by western blotting. Signals were detected with SuperSignalWest Pico (Pierce, Rockford, IL, USA), and the immunoreactive bands were visualized by LAS-4000 Luminescent Image Analyser (FUJIFILM). The intensity of band was measured using LabWorks Version 4.5 software (UVP, LLC). The following antibodies were used in this study at indicated dilutions: anti-DYKDDDDK tag (Wako (Japan), catalog #012-22384, clone 1E6, 1:2000), anti-HA (Roche, catalog #11867431001, clone 3F10, 1:2000), anti-MYC (Roche, catalog #11814150001, clone 9E10, 1:1000), anti-GAPDH (Cell Signaling Technology, catalog #5174, clone D16H11, 1:1000), anti-α-tubulin (Sigma-Aldrich, catalog #T9026, clone DM1A, 1:3000), anti-ubiquitin (Cell Signaling Technology, catalog #3936, clone P4D1, 1:500), anti-ASXL1 (Santa Cruz Biotechnology, catalog #sc-81053, clone 2049C2a, 1:1000), anti-UBE2O polyclonal antibody (Cell Signaling Technology, catalog #83393S, Lot 1, 1:1000), anti-ubiquityl-histone H2A (Lys119) (Cell Signaling Technology, catalog #8240, clone D27C4, 1:2000), anti-BAP1 antibody (Santa Cruz Biotechnology, #sc-28383, clone C-4, 1:1000), anti-HOXA5 (Santa Cruz Biotechnology, catalog #sc-365784, clone C-11, 1:200), anti-HOXA7 polyclonal antibody (Millipore, catalog #09-086, Lot 3022838, 1:500), anti-HOXA9 polyclonal antibody (Millipore, catalog 07-086, Lot 2899748, 1:500), anti-IRF8 (Cell Signaling Technology, catalog #5628, clone D20D8, 1:500). Anti-histone H3 and anti-H3K27me3 were kind gifts from Dr. Kimura[63]. Uncropped blots are provided in Supplementary Figs. 11–14.

**Quantitative RT-PCR**. Total RNA was extracted using the RNeasy Mini kit (QIAGEN). RNA was reverse transcribed using Omniscript RT kit (Qiagen). Complementary DNA (cDNA) was then subjected to quantitative RT-PCR using a SYBR Select Master Mix (Applied Biosystems). Sequences of the primers used for qRT-PCR in this study, from 5′ to 3′ are as follows: AGGTAGCGGTTGAAG TGGAA (mouse Hoxa5 Forward), CGCAAGCTGCACATTAGTCA (mouse Hoxa5 Reverse), AGGTAGCGGTTGAAATGGAA (mouse Hoxa7 Forward), GAAGCCAGTTTCCGCATCTA (mouse Hoxa7 Reverse), GTTCCAGCGTC TGGTGTTTT (mouse Hoxa9 Forward), ACAATGCCGAGAATGAGAGC (mouse Hoxa9 Reverse), CGTGGAAGACGAGGTTACGCTG (mouse Irf8 Forward), GCTGAATGGTGTGTGTCATAGGC (mouse Irf8 Reverse), AATTCCCAGTCAG AGGCGCTATGT (mouse Pten Forward), GATTGCAAGTTCCGCCACTGAACA (mouse Pten Reverse), GTGGGGCCTTGATTCCACAGG (mouse Sf3b1 Forward), GGCTTCTTCTGACCGAGCAA (mouse Sf3b1 Reverse), TTGATGGCAACAATC

**Fig. 9** BAP1 depletion inhibits the growth of myeloid leukemia cells with ASXL1 mutations or MLL-fusions. **a–c** Murine bone marrow c-Kit$^+$ cells were transformed by MLL-AF9. The MLL-AF9 leukemia cells were transduced with Cas9 and Bap1-targeting sgRNAs, and were cultured in semisolid medium or transplanted into recipient mice. **a** Vector or human BAP1-transduced (tdTomato$^+$) cells were sorted, and were then transduced with vector or two independent sgRNAs targeting mouse Bap1 (sgBap1 #1, #2). Colony-forming activity was assessed using these cells. Shown are the colony counts per $10^4$ plated cells (mean ± s.e.m.) from quadricate plates. Expression of human BAP reversed the reduced colony formation of Bap1-depleted MLL-AF9 cells. ****$P < 0.0001$, two-way ANOVA with Tukey's multiple comparison test. **b** Survival curves of recipient mice transplanted with vector- or sgBAP1-transduced MLL-AF9 cells ($n = 7$ for each group). **$P = 0.0010$, $^{\$\$}P = 0.0026$, log-rank test. **c** Relative expression of Hoxa5, Hoxa7, Hoxa9, Irf8, Pten, and Sf3b1 were assessed in vector- or sgBAP1-transduced MLL-AF9 cells using the qPCR analyses. Results were normalized to Gapdh, with the relative mRNA level in vector-transduced cells set at 1. Data are shown as mean ± s.e.m. of four biological independent experiments. *$P < 0.05$, $^{\#}P < 0.01$, one-way ANOVA with Dunnett's multiple comparisons test. **d** Human AML cell lines harboring an ASXL1 mutation (Kasumi-1 and MEG-01), a CML cell line with an ASXL1 mutation (TS9;22), AML cell lines with MLL-AF9 (MOLM-13 and THP-1), a T cell leukemia cell line (Jurkat) and a B cell lymphoma cell line (Raji) were transduced with Cas9 together with a vector or two independent sgRNAs targeting human BAP1 (sgBAP1 #3 and #4). Shown are cell numbers in each culture counted on days 2, 4, and 7. BAP1-targeting sgRNAs induced efficient depletion of BAP1 in all these cells. Two independent experiments were performed, and data are shown as mean ± s.e.m. n.s. not significant, **$P < 0.01$, ***$P < 0.001$; vector versus sgBAP1#3, $^{\$\$\$}P < 0.001$; vector versus sgBAP1#4, one-way ANOVA with Dunnett's multiple comparisons test

TCCAC (mouse *Gapdh* Forward), CGTCCCGTAGACAAAATGGT (mouse *Gapdh* Reverse), TGGAACTCCTTCTCCAGCT (human *HOXA5* Forward), AGATCTACCCCTGGATGCG (human *HOXA5* Reverse), CGTCAGGTAGCGG TTGAAGT (human *HOXA7* Forward), AATTTCCGCATCTACCCCTG (human *HOXA7* Reverse), CAGTTCCAGGGTCTGGTGTT (human *HOXA9* Forward), AATGCTGAGAATGAGAGCGG (human *HOXA9* Reverse), TTCCCTTT AAAAACTGCCCA (human *IRF8* Forward), CCAGGACTGATTTGGGAGAA (human *IRF8* Reverse), TGGATTCGACTTAGACTTGACCT (human *PTEN* Forward), GCGGTGTCATAATGTCTCTCAG (human *PTEN* Reverse), GTGGGCCTCGATTCTACAGG (human *SF3B1* Forward), GATGTCACGTAT CCAGCAAATCT (human *SF3B1* Reverse), CTCAGGGGTGAATTCTTTGC (human *HBG1* Forward), GTGACAAGCTGCATGTGGAT (human *HBG1* Reverse), GCTGGTCCTCAGACTTCACG (human *KLF1* Forward), CACACAG GATGACTTCCTCAA (human *KLF1* Reverse), TTGAGGTCAATGAAGGGGTC (human *GAPDH* Forward), GAAGGTGAAGGTCGGAGTCA (human *GAPDH* Reverse).

**ChIP-qPCR assay**. ChIP was performed using Simple chip kit (Cell Signaling Technology, #9002) with antibodies against H2AK119 (Cell Signaling Technology, catalog #8240, 1:100) and HA (abcam, catalog #ab9110, 1:100), following the manufacturer's recommendations. Purified DNA was then subjected to quantitative RT-PCR using a SYBR Select Master Mix (Applied Biosystems). Sequences of the primers used for ChIP-qPCR in this study, from 5′ to 3′, are as follows: CTCCCCCGAATCCTCTGTAT (mouse *Hoxa5* Forward), GGGGTCGAATTG AGGTTACA (mouse *Hoxa5* Reverse), CTGTGAGGGCTGCTGAGATT (mouse *Hoxa7* Forward), GCAGCTTTCAGTGTCGGTTT (mouse *Hoxa7* Reverse), AGTACAGAGGCAAGGCCAGA (mouse *Hoxa9* Forward), CCGGTGTTT TGCAGTCATAA (mouse *Hoxa9* Reverse), AACAGCCTTTCGGTTTTCCT (mouse *Irf8* Forward), ACTGCCGGAAGATCAGAGGT (mouse *Irf8* Reverse), TGCGAGGATTATCCGTCTTC (mouse *Pten* Forward), GCTGGATGGTTGCAG AGACT (mouse *Pten* Reverse), GAGAGCGGCCTCTGTTTCT (mouse *Sf3b1* Forward), ATCTTCGCCATTTTGTCCAC (mousee *Sf3b1* Reverse).

**RNA-seq analysis**. Total RNA was extracted using the RNeasy Mini Kit (Qiagen). RNA integrity was examined with the TapeStation (Agilent Technologies). cDNA was prepared from polyA-selected RNA by using an NEBNext Ultra Directional RNA Library Prep kit (New England BioLabs) and was subjected to next-generation sequencing from both ends with the HiSeq 2500 platform (Illumina). For expression profiling with RNA-seq data, paired-end reads were aligned to the mm10 mouse genome assembly using the TopHat2 computational pipeline (https://ccb.jhu.edu/software/tophat/index.shtml). The expression level for each gene was calculated from mapped read counts using HTSeq (http://www-huber. embl.de/users/anders/HTSeq/doc/overview.html) and normalized with the DESeq2 pipeline (http://bioconductor.org/packages/release/bioc/html/DESeq2.html). For clustering analysis, normalized read counts were further transformed by the variance-stabilizing transformation method in DESeq2. They were then subjected to hierarchical clustering analysis with Ward's method.

**Nano-LC/MS MS**. 293T cells were transfected with FLAG-ASXL1-MT together with HA-BAP1 or control vector. Forty-eight hours after transfection, whole-cell extracts were incubated with an anti-FLAG M2 Affinity Gel for 2 h at 4 °C. Affinity gels were stringently washed, and then proteins bound to the affinity gel were eluted in buffer containing 0.5 mg/ml FLAG peptide for 30 min at 4 °C. After the gel was removed by centrifugation, the supernatants were filtered by using an Ultrafree-MC filter (Millipore). A 10% (vol/vol) volume of the eluted proteins that co-precipitated with ASXL1-MT was subjected to SDS/PAGE, followed by silver staining. The remaining 90% (vol/vol) volume of eluted proteins was trypsinized and subjected to nano liquid chromatography tandem mass spectrometry (nano-LC-MS/MS) analysis to identify co-immunoprecipitated host proteins. For this analysis, we used LTQ-Orbitrap Velos (Thermo Fisher Scientific) coupled with Dina-2A (KYA Technologies). The MS/MS signals were processed against the RefSeq (National Center for Biotechnology Information) human protein database (35,853 sequences as of February 4, 2013) using the Mascot algorithm (version 2.5.1; Matrix Science). Carbamidomethylation of cysteine was set as a fixed modification, whereas oxidation (Met), protein N-terminal acetylation, pyro-glutamination (Gln), phosphorylation (Ser, Thr, and Tyr), and diglycine (Lys) were set as variable modifications. Trypsin was defined as a proteolytic enzyme and a maximum of two missed cleavages were allowed. The mass tolerance was set to three parts per million (ppm) for peptide masses and 0.8 Da for MS/MS peaks. In the process of peptide identification, we conducted decoy database searching by Mascot and applied a filter to satisfy a false-positive rate lower than 1%.

**Statistics**. Statistical analyses were performed by the unpaired and two-tailed Student's *t* test or by analysis of variance (ANOVA) with Dunnett's or Turkey's post hoc test after testing for normal distribution and equal variance. The survival distributions were compared by the log-rank test. GraphPad Prism 7 was used for these statistical analyses. Samples sizes were estimated by preliminary experiments. No specific statistical methods were used to predetermine the sample size.

**Study approval**. All animal experiments were approved by the Animal Care Committee at the Institute of Medical Science, the University of Tokyo (approval number: K15-32, PA13-19, and A17-75). All experiments using human cord blood cells were approved by the Ethics Committee at the Institute of Medical Science, the University of Tokyo (approval number: 27-34-1225).

**Data availability**. RNA-seq data from this study have been deposited into the NCBI Gene Expression Omnibus (GEO) under accession number GSE114861 [https://www.ncbi.nlm.nih.gov/geo/query/acc.cgi?acc=GSE114861]. The data that support the findings of this study are available from the authors upon request.

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

## Acknowledgements

We are grateful to Dr. Yuichi J. Machida for providing plasmids regarding BAP1 constructs. We are grateful to Dr. Peter ten Dijke for providing plasmids regarding UBE2O constructs. We are grateful to Dr. Hiroshi Kimura for the kind gifts of the H3 and H3K27me3 antibodies. We thank the FACS Core, the Mouse Core and the Microscopy Core Laboratories at the Institute of Medical Science, the University of Tokyo. This work was supported by a Grant-in-Aid Scientific Research B from the Ministry of Education, Culture, Sports, Science and Technology of Japan (15H04855, T.K.), a grant from the Tokyo Biochemical Research Foundation (T.K.), a grant from the Uehara Memorial Foundation (T.K.), a grant from The Japan foundation for Aging and Health (S.G.), a grant from The Cell Science Research Foundation (S.G.), a grant from Suzuken Memorial Foundation (S.G.), and a grant from The SENSHIN Medial Research Foundation (S.G.). We are grateful to Dr. Douglas Sipp for excellent language support.

## Author contributions

S.A. designed and performed most of the experiments, analyzed and interpreted the data, and wrote the manuscript. S.G. conceived the project, designed experiments, interpreted the data, and wrote the manuscript. S.S., R.T., Tsu.F., T.Y., and Ta.F. assisted in the experiments. D.I., A.Y., S.Y., Y.H., K.C.K., To.F., and Y.T. advised on data interpretation. H.K.-H. and M.O. performed the experiments involving nano-LC/MS MS. S.K., M.K., and H.M. performed the RNA-seq experiments. T.K. conceived the project, interpreted the data, and participated in writing the manuscript.

## Additional information

**Competing interests:** The authors declare no competing interests.

