## [peer Review File · Nature Communications]

Reviewer #1 (Remarks to the Author):

The manuscript by Asada et al makes the major claim that a c-terminal truncated form of ASXL1 complexes and cooperates with BAP1 in survival and proliferation of myeloid leukemia cells. This adds novel information to the field regarding the function of mutant ASXL1 as well as provides novel data indicating that BAP1 can participate in promoting myeloid leukemia growth (rather than a sole role as a tumor suppressor). The following questions/concerns arose during in-depth review of the manuscript:

Data interpretation related to Figures 3&4 was not appropriately resolved or discussed as follows:

- How do the authors interpret the findings that ASXL1-MUT and BAP1 overexpression impairs self-renewal (measured by % CD34+ cells in Figure 3F) with increased serial colony replating (Figure 4A), typically thought of as a surrogate in vitro assay to measure stem/progenitor cell self-renewal? This is also true in measurement of CD34+ frequency in supplementary Figure 4F. Is this possibly a difference between mouse and human cell phenotypes?

Some additional replicates and analyses are needed to support the reproducibility of the work, as follows:

- Figure 1G: can this figure be annotated with relative density of bands (FLAG/a-tubulin) as was done in Figure 2A? This would clarify the increased intensity of the FLAG bands with sgUBE2O.

- Figure 2C: to quantify subcellular localization of BAP1, how many cells were counted to generate the bar graph?

- Figure 5B: all genes shown in Figure 5B, D, E, F are shown to be increased in expression, ASXL1-MT occupancy and have decreased H2AK119ub. To demonstrate that this is not due to a technical issue with this experiment, the authors should include a negative control gene demonstrating no change in expression, ASXL1-MT occupancy or H2AK119ub.

- Figure 5B: clarify whether the n=3 is referring to technical or biological replicates

- Figure 5F: data presented from triplicate wells, it is unclear if this experiment was repeated more than once in its entirety, which is critical to demonstrate reproducibility. Statistical analysis should also be performed

- Figure 6H: same comment as Figure 5F

In addition, some of the language used in the paper is inconsistent with the results, as follows:

- results paragraph 1 and Figure 1 is testing "overexpression" of BAP1 not "presence" of BAP1. The latter would be most appropriately tested by knockout of BAP1.

- results describing Figure 2C state that "nuclear import of BAP1 is not observed with ASXL1-MT-K351R". The data in the figure show that nuclear import is observed, although to a lesser extent than that of the ASXL1-MT.

Addressing the above comments/concerns would substantially improve the rigor and reproducibility, and support the major claims of the paper.

Sincerely,

Jennifer Trowbridge

Reviewer #2 (Remarks to the Author):

ASXL1 is frequently mutated in myeloid malignancies. The underlying mechanisms by which mutant ASXL1 mediated leukemogenesis remain largely unclear. While the ASXH domain has been shown to interact with BAP1, polycomb repressive deubiquinase, the impact of the changes in the PR-DUB activity in the ASXL1 mutation-related myeloid pathogenesis is unknown. In the current study, Asada and colleagues report a mutually reinforcing effect between mutant ASXL1 and BAP1 in promoting myeloid malignancies. They found that BAP1 stabilizes and monoubiquitinates truncated ASXL1, leading to the alteration of multi-lineage differentiation of haematopoietic progenitors and accelerating RUNX1-ETO associated leukaemogenesis. They further demonstrated that reducing BAP1 activity by CRISPR/Cas9 inhibited the pathogenesis of ASXL1 mutation mediated myeloid leukaemia. This study provides a strong insight that besides a tumor suppressor effect, BAP1 also plays a tumor promoting role myeloid malignancies. This study is well designed and with significant clinical relevance. The paper is well written. The reviewer's comments are described as following:

1. In Figure 1, the authors showed that BAP1 stabilizes ASXL1-MT and induces its monoubiquitination at lysine 351. While the mechanism remains unknown, future work to identify the related mechanism should be commented in the discussion. The sizes for figure 1b, c, d and g should be labeled. In addition, it is not clear how many times the experiments were repeated for the studies in Figure 1.

2. Figure 2, the authors nicely showed that monoubiquitinated ASXL1-MT enhances BAP1 catalytic activity. The sizes for figure 2a, d, e and f should be labeled. Please provide the “n” value for each of the experiments. Figure 2c, please clarify the total cell number assessed.
3. In Figure 3, the authors analyzed the effect of ASXL1-MT on mast cell differentiation. Clearly, the ASXL1-MT alone modestly impaired mast cell maturation. Since the cells used for this assay is c, the word “HSPCs” here should be replaced by “murine myeloid cell line”. In addition, the following words “which was augmented by BAP1 coexpression” is confusing, and “augmented” should be replaced or changed to reflect that the inhibitory effect was triggered by
4. Figure 4d what type of cells are these GFP+ cells? In Figure 4i, the size of the bands should be labeled.
5. Figure 5d, e, the labeling of “murine HSPC cells”, should be changed to cKit+ cells.
6. There are some miss used notations for gene and proteins, e.g. small cases, vs. capital cases.
7. Since some of the studies contain more than two genotypes of cells, student t-test is not appropriated for statistical analyses. Instead, ANOVA may apply for the statistical analyses.

Reviewer #3 (Remarks to the Author):

In this study, Asada and colleagues investigated the molecular mechanism of a C-terminally truncated mutant of ASXL1 (ASXL1-MT) and BAP1 in promoting myeloid leukaemogenesis. They found that BAP1 induces monoubiquitination of ASXL1-MT at K351. This, in turn, enhances the catalytic activity of BAP1 toward histone H2AK119Ub, leading to aberrant myeloid differentiation in hematopoietic progenitor cells and increased leukaemogenesis in a RUNX1-ETO-driven model. The authors then provide experimental evidence by RNAseq, qPCR and ChIP-qPCR to demonstrate that the hyperactive ASXL1-MT/BAP1 complex directly regulates expression of posterior HOXA genes and IRF8. This is achieved by removing the monoubiquitin modification of histone H2AK119. It is proposed that this is the potential mechanism by which ASXL1-MT/BAP1's promotes leukaemogenesis. Their data, therefore, suggest a tumor-promoting role of BAP1 in myeloid

neoplasms, despite its well-defined tumor suppressor activity in other tumor models. This study has some interesting observations, such as the mutually reinforcing model between ASXL1-MT and BAP1 through monoubiquitination at K351, and the direct regulation of HOXA genes and IRF8 in multiple myeloid neoplasms. However, many of the observations are disconnected, and lack compelling experimental evidence to establish a causal relationship.

Major concerns:

1. It is interesting to learn that expression of BAP1, a deubiquitinating enzyme, promotes rather than reduces the monoubiquitination of ASXL1-MT at K351. Though the authors claim this is achieved through BAP1 recruiting an E3 ligase complex, their failure to identify such a complex undermines this assertion. Why does BAP1 promote monoubiquitination of ASXL1-MT, but not full length ASXL1? It is important to point out that this monoubiquitination site (K351) is located within the DEUBAD domain, essential for BAP1 binding (Sahtoe et al., 2015). Hence, it will be important to compare the binding affinity of BAP1 for ASXL1-MT versus ASXL1-MT K351R, as the reduced activity of K351R may well be due to reduced ASXL1/BAP1 interaction - not loss of monoubiquitination. Also, the ASXL1-MT mutant may possess higher affinity to BAP1 compared to the full-length ASXL1 protein, explaining its hyperactivity.

2. The authors claim that the monoubiquitination of ASXL1-MT enhances the catalytic function of BAP1, and it does so by enhancing the autodeubiquitination activity of BAP1 and increasing its nuclear retention. However, a key control experiment is missing. Per this hypothesis, the ASXL1-MT and ASXL1-MT K351R should show no difference in promoting the nuclear retention of BAP1 C91A mutant. This control should be added to this study.

3. In the RUNX1-ETO-induced leukaemogenesis model and the c-Kit⁺ cells, the authors observed increased transcription of HOXA genes and IRF8, and conclude that the dysregulation of these genes is responsible for the transforming ability of the ASXL1-MT/BAP1 complex. However, this is merely an association rather than a causal relationship. More experimental evidence is needed to cement this conclusion. First, the authors should determine the protein level of these genes by western blot, and CRISPR knockout the candidate genes in ASXL1-MT/BAP1 expressing cells to see if this counters transforming or differentiation events.

4. The authors showed that in the RUNX1-ETO CB cell model (i) the expression of ASXL1-MT alone is sufficient to promote the growth of cells, and (ii) that endogenous levels of BAP1 are sufficient for inhibiting ASXL1-MT driven differentiation. In Figure 1b, 1g and 1h, however, they clearly show that expression of ASXL1-MT alone doesn't result in a monoubiquitination-mediated

shift, indicating that endogenous BAP1 is, in fact, not sufficient. Is there some cell or model specificity?

Some Minor concerns:

1. Majority of ASXL1 mutations identified in myeloid neoplasms result in C-terminally truncated proteins. It will be interesting to test multiple mutations in their key assays, such as the monoubiquitination-caused shift of ASXL1, or protein stability with or without BAP1. These data will highlight the generality of this model.
2. What is the molecular or cellular basis of enhanced ASXL1-MT/BAP1 catalytic activity? Does it increase the enzymatic activity of the complex, or does it promote the recruitment of this complex to chromatin?
3. In Figure 1c and 1d, it will be interesting to see whether co-expression of BAP1 stabilizes ASXL1-MT K351R. This will indicate that monoubiquitination plays a role in ASXL1 stability.
4. This study is based on over-expression of ASXL1-MT, in the cell lines that harbor ASXL1 mutation, such as Kasumi-1, MEG-01 and TS9;22. Does endogenous ASXL1 show a monoubiquitination-mediated shift?
5. The authors state that interaction between ASXL1-MT and the atypical E2/E3 hybrid ligase UBE2O occur only in cells where BAP1 is coexpressed. If so, then why in Figure 1f is an interaction detected between UBE2O and ASXL1-MT in the absence of BAP1 expression?

** See Nature Research's author and referees' website at www.nature.com/authors for information about policies, services and author benefits

To Reviewer #1's comments

[Comment #1]

Data interpretation related to Figures 3&4 was not appropriately resolved or discussed as follows:

- How do the authors interpret the findings that ASXL1-MUT and BAP1 overexpression impairs self-renewal (measured by % CD34+ cells in Figure 3F) with increased serial colony replating (Figure 4A), typically thought of as a surrogate in vitro assay to measure stem/progenitor cell self-renewal? This is also true in measurement of CD34+ frequency in supplementary Figure 4F. Is this possibly a difference between mouse and human cell phenotypes?

[RESPONSE to Comment #1]

Thank you for the suggestion. Several oncogenes with the ability to enhance self-renewal of HSCs, such as RUNX1-ETO, were shown to increase CD34 expression in cord blood cells. In contrast, coexpression of ASXL1-MT and BAP1 reduced the frequency of CD34 expression in cord blood cells and those expressing RUNX1-ETO. Based on the observation, we stated that mutant ASXL1 and BAP1 impaired self-renewal of HSPCs. However, we now realize that CD34 expression does not necessarily surrogate the self-renewal capacity of stem/progenitor cells (Some oncogenes, such as MLL-AF9, increases self-renewal capacity of HSPCs but does not increase CD34 expression). Therefore, we decided to simply describe the observed results in the revised manuscript, as follows: "combination of ASXL1-MT and BAP1 did not simply increase the frequency of HSPCs" (Page 11, line 231-232).

[Comment #2]

Some additional replicates and analyses are needed to support the reproducibility of the work, as follows: - Figure 1G: can this figure be annotated with relative density of bands (FLAG/a-tubulin) as was done in Figure 2A? This would clarify the increased intensity of the FLAG bands with sgUBE2O.

[RESPONSE to Comment #2]

We quantified the relative density of bands (FLAG/a-tubulin) and confirmed that UBE2O depletion stabilized ASXL1-MT protein. We have displayed the data in original Figure 1g (corresponding to new Figure 2d).

[Comment #3]

- Figure 2C: to quantify subcellular localization of BAP1, how many cells were counted to generate the bar graph?

[RESPONSE to Comment #3]

We counted 400 cells for each culture. We have added this information to figure legends of original Figure 2c (corresponding to new Figure 3c) in the revised manuscript.

[Comment #4]

- Figure 5B: all genes shown in Figure 5B, D, E, F are shown to be increased in expression, ASXL1-MT occupancy and have decreased H2AK119ub. To demonstrate that this is not due to a technical issue with this experiment, the authors should include a negative control gene demonstrating no change in expression, ASXL1-MT occupancy or H2AK119ub.

[RESPONSE to Comment #4]

Thank you very much for your valuable comment. In response to the comment, we assessed ASXL1-MT binding and H2AK119ub on promoter loci of *Pten* and *Sf3b1*, because neither overexpression of ASXL1-MT/BAP1 (original Figure 5f, corresponding to new Figure 6f) nor Bap1 depletion (original Figure 6h, 7c corresponding to new Figure 7h, 9c, respectively) changed their expression in cells. As expected, ASXL1-MT occupancy as well as the level of H2AK119ub were not significantly altered at these regions (original Figure 5d, e corresponding to New Figure 6d, e). We have added these data to the revised manuscript, and have modified text accordingly (Page 15, line 309-311).

[Comment #5]

- Figure 5B: clarify whether the n=3 is referring to technical or biological replicates

[RESPONSE to Comment #5]

We performed three biologically independent experiments. We have added the explanation to the Figure legend of original Figure 5b (corresponding to new Figure 6b) in the revised manuscript.

[Comment #6]

- Figure 5F: data presented from triplicate wells, it is unclear if this experiment was repeated more than once in its entirety, which is critical to demonstrate reproducibility. Statistical analysis should also be performed

- Figure 6H: same comment as Figure 5F

[RESPONSE to Comment #6]

According to the reviewer's comment, we repeated the experiments four times in total, and performed statistical analyses. We have added the data (original Figure 5f, 6h corresponding to new Figure 6f and 7h, respectively) to the revised manuscript.

[Comment #7]

In addition, some of the language used in the paper is inconsistent with the results, as follows:

- results paragraph 1 and Figure 1 is testing "overexpression" of BAP1 not "presence" of BAP1. The latter would be most appropriately tested by knockout of BAP1.

[RESPONSE to Comment #7]

Thank you very much for the thoughtful comment. In revised manuscript, we have performed new experiments using BAP1-targeting sgRNAs and showed that endogenous BAP1 is indeed essential for monoubiquitination of ASXL1-MT. We have now added the data to the revised manuscript (New Figure 1h, Supplementary Figure 1d), and have rephrased the sentences accordingly to “Expression of BAP1 stablized mutant ASXL1 and induces its monoubiquitination at lysine 351” (Page5, line 105-106).

[Comment #8]

- results describing Figure 2C state that "nuclear import of BAP1 is not observed with ASXL1-MT-K351R". The data in the figure show that nuclear import is observed, although to a lesser extent than that of the ASXL1-MT.

[RESPONSE to Comment #8]

Thank you for the important comment. According to your suggestion, we have revised the statement to “observed only weakly when we used the ubiquitination-deficient ASXL1-MT-K351R” (Page 9, line 180-181).

To Reviewer #2's comments

[Comment #1]

In Figure 1, the authors showed that BAP1 stabilizes ASXL1-MT and induces its monoubiquitination at lysine 351. While the mechanism remains unknown, future work to identify the related mechanism should be commented in the discussion. The sizes for figure 1b, c, d and g should be labeled. In addition, it is not clear how many times the experiments were repeated for the studies in Figure 1.

[RESPONSE to Comment #1]

Thank you for raising an important point. As the reviewer pointed it out, deciphering mechanisms of BAP1-induced monoubiquitination of ASXL1-MT is an important future challenge. To identify the specific ubiquitin ligase to mediate BAP1-induced ASXL1-MT's monoubiquitination, we depleted 20 candidate E3 ubiquitin ligases including BAP1 interacting proteins, BRCA1 (PMID:9528852) and BARD1(PMID:19117993) in 293T cells coexpressing ASXL1-MT and BAP1. However, depletion of any individual ligases did not reduce the monoubiquitination of ASXL1-MT (data not shown). These data strongly indicate the involvement of multiple E3 ligases in this process. We have added the explanation to the revised manuscript (Page 22, line 461-467).

For original Figure 1b, c, d, and g (corresponding to new Figure 1b, c, d and 2d, respectively), we added the size of the molecular weight markers. We performed at least three times for the data shown in Figure 1. We have added this explanation to the figure legend.

[Comment #2]

Figure 2, the authors nicely showed that monoubiquitinated ASXL1-MT enhances BAP1 catalytic activity. The sizes for figure 2a, d, e and f should be labeled. Please provide the “n” value for each of the experiments. Figure 2c, please clarify the total cell number assessed.

[RESPONSE to Comment #2]

We have added the sizes of the molecular weight markers to original Figure 2a, d, e and f (corresponding to new Figure 3a, d, e and f). We analyzed 400 cells to produce the graph displayed in original Figure 2c (corresponding to new Figure 3c). We performed each experiment at least three times. We have revised figure legends accordingly.

[Comment #3]

In Figure 3, the authors analyzed the effect of ASXL1-MT on mast cell differentiation. Clearly, the ASXL1-MT alone modestly impaired mast cell maturation. Since the cells used for this assay is c, the word “HSPCs” here should be replaced by “murine myeloid cell line”. In addition, the following words “which was augmented by BAP1 coexpression” is confusing, and “augmented” should be replaced or changed to reflect that the inhibitory effect was trigger by

[RESPONSE to Comment #3]

We used murine c-kit⁺ progenitor cells in experiments shown in original figure 3a, b and c (corresponding to New Figure 4a, b and c). We have revised the statement from “HSPCs” to “c-kit⁺ progenitor cells” according to the reviewer’s suggestion (Page 10, line 205). We have also revised the statement “which was augmented by BAP1 coexpression” to “this inhibitory effect was enhanced by BAP1 coexpression” (Page 10, line 207-208). Thank you very much for pointing these out.

[Comment #4]

Figure 4d what type of cells are these GFP⁺ cells? In Figure 4i, the size of the bands should be labeled.

[RESPONSE to Comment #4]

“GFP⁺ cells” indicate the cells transduced with vector, ASXL1-MT, or ASXL1-MT K351R. We have added the information to the figure legend. We have also added the sizes of the molecular weight markers to original Figure 4i (corresponding to New Figure 5i).

[Comment #5]

Figure 5d, e, the labeling of “murine HSPC cells”, should be changed to cKit⁺ cells.

[RESPONSE to Comment #5]

In response to the reviewer’s comment, we have changed the labeling in original Figure 5d, e (corresponding to new Figure 6d, e). Thank you very much for raising this point.

[Comment #6]

There are some miss used notations for gene and proteins, e.g. small cases, vs. capital cases.

[RESPONSE to Comment #6]

We have revised the text according to the reviewer’s suggestion. Thank you very much.

[Comment #7]

Since some of the studies contain more than two genotypes of cells, student t-test is not appropriated for statistical analyses. Instead, ANOVA may apply for the statistical analyses.

[RESPONSE to Comment #7]

We thank the reviewer’s thoughtful suggestion. We re-analyzed our data using ANOVA (new Figure 4a-f, 5a, 5e, 5f, 6a, 6f, 7b, 7f, 7g, 9a, 9c and 9d) and have changed figure legends accordingly. The new analyses did not change our conclusion.

To Reviewer #3's comments

[Comment Major #1-1]

It is interesting to learn that expression of BAP1, a deubiquitinating enzyme, promotes rather than reduces the monoubiquitination of ASXL1-MT at K351. Though the authors claim this is achieved through BAP1 recruiting an E3 ligase complex, their failure to identify such a complex undermines this assertion.

[RESPONSE to Comment Major #1-1]

We completely agree with the reviewer that it is important to identify the E3 ligase responsible for the monoubiquitination of ASXL1-MT. We selected twenty ubiquitin ligases, including BAP1 interacting proteins, BRCA1 (PMID:9528852) and BARD1(PMID:19117993), as candidate E3 ligases that mediate BAP-induced monoubiquitination of ASXL1-MT. We then depleted them individually using CRISPR/Cas9 system in 293T cells coexpressing ASXL1-MT and BAP1. However, depletion of any individual ligase did not reduce the BAP1-induced monoubiquitination of ASXL1-MT (data not shown). These data strongly indicate the involvement of multiple E3 ligases with redundant functions in this process and we think that it requires further works to clarify this important issue. We have added the explanation to the revised manuscript (Page 22, line 461-467).

[Comment Major #1-2]

Why does BAP1 promote monoubiquitination of ASXL1-MT, but not full length ASXL1?

[RESPONSE to Comment Major #1-2]

To clarify this important point raised by this reviewer, we performed immunoprecipitation assay and compared the binding affinity of BAP1 for wild-type ASXL1 versus ASXL1-MT. ASXL1-MT deprived BAP1 from ASXL1-WT in a dose dependent manner. On the other hands, wild-type ASXL1 did not inhibit the interaction between BAP1 and ASXL1-MT (new Supplementary Figure 2a, b). These data suggest that BAP1 binds to ASXL1-MT more strongly than wild-type ASXL1, and it could explain why BAP1 selectively promote monoubiquitination of ASXL1-MT. We have added these data to the revised manuscript and modified the text accordingly (Page 6, line 119-120).

[Comment Major #1-3]

It is important to point out that this monoubiquitination site (K351) is located within the DEUBAD domain, essential for BAP1 binding (Sahtoe et al., 2015). Hence, it will be important to compare the binding affinity of BAP1 for ASXL1-MT versus ASXL1-MT K351R, as the reduced activity of K351R may well be due to reduced ASXL1/BAP1 interaction - not loss of monoubiquitination. Also, the ASXL1-MT mutant may possess higher affinity to BAP1 compared to the full-length ASXL1 protein, explaining its hyperactivity.

[RESPONSE to Comment Major #1-3]

We thank the reviewer's insightful recommendations and agree that these are important points. To compare the binding affinity of BAP1 for ASXL1-MT versus ASXL1-MT K351R, we used FLAG/MYC tagged constructs and performed competitive immunoprecipitation assay. This assay revealed that both ASXL1-MT and ASXL1-MT-K351R bind to BAP1 with similar affinities. We have added these data to the revised manuscript (new Supplementary Figure 2c, d), and modified the text accordingly (Page 6, line 120-124).

[Comment Major #2]

The authors claim that the monoubiquitination of ASXL1-MT enhances the catalytic function of BAP1, and it does so by enhancing the autodeubiquitination activity of BAP1 and increasing its nuclear retention. However, a key control experiment is missing. Per this hypothesis, the ASXL1-MT and ASXL-MT K351R should show no difference in promoting the nuclear retention of BAP1 C91A mutant. This control should be added to this study.

[RESPONSE to Comment Major #2]

Thank you for the important comment. In response to the reviewer's comment, we performed the immunofluorescence assay using BAP1-C91S as a control. BAP1 C91S coexpressed with vector control have not been accumulated to nucleus compared with BAP1 WT. Interestingly, BAP1 C91S still showed some nuclear localization in a fraction of cells when ASXL1-MT but not ASXL1-MT K351R was coexpressed (new Supplementary Fig.3a), probably because BAP1-C91S retains weak deubiquitinase activity and induces monoubiquitination of ASXL1-MT as shown in Figure 1b. We have added these data to the revised manuscript and modified the text accordingly (Page 9, line 181-183).

[Comment Major #3]

In the RUNX1-ETO-induced leukaemogenesis model and the c-Kit⁺ cells, the authors observed increased transcription of HOXA genes and IRF8, and conclude that the dysregulation of these genes is responsible for the transforming ability of the ASXL1-MT/BAP1 complex. However, this is merely an association rather than a causal relationship. More experimental evidence is needed to cement this conclusion. First, the authors should determine the protein level of these genes by western blot, and CRISPR knockout the candidate genes in ASXL1-MT/BAP1 expressing cells to see if this counters transforming or differentiation events.

[RESPONSE to Comment Major #3]

Thank you very much for the important suggestion. We assessed protein levels of Hoxa5, Hoxa7 Hoxa9 and Irf8 in murine c-kit positive cells transduced with control vector or ASXL1-MT/BAP1, and confirmed that ASXL1-MT and BAP1 upregulated these genes at protein levels (new Supplementary Figure 6a). We have modified the text accordingly (Page 14, line 301-304 and Page 15, Line 312-314).

To assess the role of Hox genes in ASXL1-MT/BAP1-induced leukemogenesis, we first depleted Hoxa5, Hoxa7 or Hoxa9 in cSAM cells using CRISPR/Cas9 system and assessed their colony forming ability. Depletion

of these individual Hox genes partially decreased the colony forming activity of cSAM cells (New Figure 8a, Supplementary Figure 8a). We then examined if ectopic expression of Hox genes in cSAM cells can reverse the inhibitory effect of BAP1 depletion on colony formation. As shown in New Figure 8c, d, ectopic expression of HOXA7 and HOXA9 partially rescued the reduction of colony-forming activity of Bap1-depleted cSAM cells.

Next, we transduced RUNX1-ETO9a, ASXL1-MT and BAP1 into bone marrow cells derived from Rosa26-LSL-Cas9 knockin mouse (PMID:25263330) to establish Cas9⁺ leukemia cells (new Supplementary Figure 8b). We depleted Hoxa5, Hoxa7, or Hoxa9 in RUNX1-ETO9a leukemia cells coexpressing ASXL1-MT and BAP1, and found that depletion of these individual Hox genes resulted in the partial reduction of their colony forming capacity (new Figure 8b). Taken together, these results suggest that posterior Hoxa genes contribute to leukemogenesis induced by ASXL1-MT and BAP1.

Finally, we assessed the role of Irf8 on monopoiesis induced by ASXL1-MT and BAP1. For this purpose, we purified c-kit positive bone marrow cells from Rosa26-LSL-Cas9 knockin mouse, and transduced ASXL1-MT, BAP1 and sgRNAs targeting *Irf8* into them. These cells were cultured with cytokines to promote myeloid differentiation. Irf8 depletion reduced the frequency of monocytes whereas restored that granulocytes in ASXL1-MT/BAP1 expressing cells (new Figure 8e, f and Supplementary Figure 8c, d). These data suggest that Irf8 contributes to monopoiesis induced by ASXL1-MT and BAP1. We have added these results to the revised manuscript accordingly and modified the text (Page 17-18, line 364-387).

[Comment Major #4]

The authors showed that in the RUNX1-ETO CB cell model (i) the expression of ASXL1-MT alone is sufficient to promote the growth of cells, and (ii) that endogenous levels of BAP1 are sufficient for inhibiting ASXL1-MT driven differentiation. In Figure 1b, 1g and 1h, however, they clearly show that expression of ASXL1-MT alone doesn't result in a monoubiquitination-mediated shift, indicating that endogenous BAP1 is, in fact, not sufficient. Is there some cell or model specificity?

[RESPONSE to Comment Major #4]

We thank the Reviewer's insightful comment and agree that it's a very important point. For experiments shown in original Figure 1b, 1g and 1h (corresponding to new Figure 1b, 2d and 2e, respectively), we transfected ASXL1-MT and/or BAP1 into 293T cells. Because the transient transfection produced high levels of ASXL1-MT, we speculate that endogenous BAP1 was not sufficient to induce monoubiquitination of the highly expressed ASXL1-MT. Therefore, we retrovirally transduced ASXL1-MT (marked with GFP) into 293T and HeLa cells to achieve lower protein level of ASXL1-MT in cells. In these cells, we detected the slowly migrating (upper) band of ASXL1-MT (monoubiquitinated ASXL1-MT) (new Figure 1g, h and Supplementary Figure 1d). Importantly, depletion of BAP1 using CRISPR/Cas9 system led to disappearance of the upper band in both 293T and HeLa cells (new Figure 1h and Supplementary Figure 1d), demonstrating the essential role of endogenous BAP1 to induce monoubiquitination of ASXL1-MT.

To further assess the relevance between the level of ASXL1-MT and BAP1-induced monoubiquitination, we then sorted GFP-high or low (expressing relatively high- or low-level of ASXL1-MT) cells, and assessed expression (and band shift) of ASXL1-MT in each fraction. Interestingly, the majority of ASXL1-MT was detected as the upper band in GFP-low fraction, while both upper and lower bands were detected in GFP-high fraction (new Figure 1f, g). These data confirm the importance of expression levels to determine the ratio of mono- and non-ubiquitinated ASXL1-MT, and suggest that the majority of endogenous mutant ASXL1 exists as the monoubiquitinated protein in cells. We have added these data to the revised manuscript, and have modified the text accordingly (Page 6-7, line 126-143).

[Comment Minor #1]

Majority of ASXL1 mutations identified in myeloid neoplasms result in C-terminally truncated proteins. It will be interesting to test multiple mutations in their key assays, such as the monoubiquitination-caused shift of ASXL1, or protein stability with or without BAP1. These data will highlight the generality of this model.

[RESPONSE to Comment Minor #1]

According to the reviewer's recommendation, we tested two additional common ASXL1 mutations (G646WfsX12 and Y591X), and confirmed that BAP1 similarly induced monoubiquitination and stabilization of them. We have added these data to the revised manuscript (new Supplemental Figure 1a), and modified the text accordingly (Page 5-6, line 110-112). Thank you very much for your valuable comment.

[Comment Minor #2]

What is the molecular or cellular basis of enhanced ASXL1-MT/BAP1 catalytic activity? Does it increase the enzymatic activity of the complex, or does it promote the recruitment of this complex to chromatin?

[RESPONSE to Comment Minor #2]

A previous report showed that deubiquitinase adaptor (DEUBAD) domain of ASXL1 increases BAP1's affinity for ubiquitin on H2A (PMID:26739236), and we showed that monoubiquitinated ASXL1-MT enhances nuclear retention of BAP1. These findings suggest that ASXL1-MT promotes the recruitment of BAP1 to chromatin, especially towards the regions with H2AK119ub. However, this concept needs to be experimentally confirmed in future research. We added this explanation to the revised manuscript (Page22, 467-472).

[Comment Minor #3]

In Figure 1c and 1d, it will be interesting to see whether co-expression of BAP1 stabilizes ASXL1-MT K351R. This will indicate that monoubiquitination plays a role in ASXL1 stability.

[RESPONSE to Comment Minor #3]

We thank the reviewer's thoughtful suggestion. In response to the reviewer's comment, we performed cycloheximide chase study in 293T cells using ASXL1-MT-K351R, and observed that BAP1 overexpression also stabilized ASXL1-MT-K351R. These data suggest that BAP1 removes polyubiquitination chains at other lysine residues from ASXL1-MT K351R. We have added these data to the revised manuscript (new Supplemental Figure 1c) and modified the text accordingly (Page 6, line 122-124).

[Comment Minor #4]

This study is based on over-expression of ASXL1-MT, in the cell lines that harbor ASXL1 mutation, such as Kasumi-1, MEG-01 and TS9;22. Does endogenous ASXL1 show a monoubiquitination-mediated shift?

[RESPONSE to Minor #4]

Unfortunately, it is difficult to detect monoubiquitination-mediated shift of endogenous ASXL1 mutants because no good antibodies are available that could efficiently detect N terminal domain of ASXL1, C-terminal truncated mutants of ASXL1. We tried to raise antibodies that could efficiently detect N terminal domain of ASXL1 in vain. Another possible strategy would be to knock-in FLAG or HA tag into the site of *ASXL1* gene in cells using CRISPR/Cas9. This will enable the detection of endogenous ASXL1.

[Comment Minor #5]

The authors state that interaction between ASXL1-MT and the atypical E2/E3 hybrid ligase UBE2O occur only in cells where BAP1 is coexpressed. If so, then why in Figure 1f is an interaction detected between UBE2O and ASXL1-MT in the absence of BAP1 expression?

[RESPONSE to Minor #5]

We added the proteasome inhibitor MG132 in experiments shown in original Figure 1f (corresponding to new Figure 2c), but not in experiments for the mass spectrometry analysis. Another difference is that we detected endogenous UBE2O in the mass spec analysis, whereas we overexpressed UBE2O in experiments for original Figure 1f (corresponding to new Figure 2c). To validate the mass spectrometry analysis data, we performed immunoprecipitation assay without MG132. This experiment revealed that ASXL1-MT interacted with UBE2O only in cells overexpressing BAP1. The data clearly indicate that BAP1 promotes the interaction between ASXL1-MT and UBE2O. We have added the data to the revised manuscript (new Supplemental Figure 1e), and modified the text accordingly (Page 7, line 152-154). Thank you very much for raising this point.

Reviewer #1 (Remarks to the Author):

The reviewers have adequately addressed the comments and critiques of the initial review. The manuscript is strengthened in both data interpretation and robustness of the experimental data presented. The findings that mutant ASXL1 cooperates with BAP1 to promote myeloid neoplasms are novel and of interest to others in the community and the wider field.

Reviewer #2 (Remarks to the Author):

The revised manuscript improved significantly. The authors have adequately address the concerns of the reviewer. No further concerns.

Reviewer #3 (Remarks to the Author):

All concerns have been satisfied.